# Sex-specific effects of cooperative breeding and colonial nesting on prosociality in corvids

Lisa Horn[1]*, Thomas Bugnyar[1], Michael Griesser[2,3,4], Marietta Hengl[1,5], Ei-Ichi Izawa[6], Tim Oortwijn[2], Christiane Rössler[1], Clara Scheer[1,7], Martina Schiestl[8], Masaki Suyama[9], Alex H Taylor[10], Lisa-Claire Vanhooland[1], Auguste MP von Bayern[11], Yvonne Zürcher[12], Jorg JM Massen[1,13]

[1]Department of Behavioral and Cognitive Biology, University of Vienna, Vienna, Austria; [2]Department of Evolutionary Biology and Environmental Studies, University of Zurich, Zurich, Switzerland; [3]Department of Biology, University of Konstanz, Konstanz, Germany; [4]Center for the Advanced Study of Collective Behaviour, University of Konstanz, Konstanz, Germany; [5]Eulen- und Greifvogelstation Haringsee, Haringsee, Austria; [6]Department of Psychology, Keio University, Tokyo, Japan; [7]Faculty of Psychology, Education and Sports, University of Regensburg, Regensburg, Germany; [8]Department of Linguistic and Cultural Evolution, Max Planck Institute for the Science of Human History, Jena, Germany; [9]Department of Behavioral Sciences, Hokkaido University, Sapporo, Japan; [10]School of Psychology, University of Auckland, Auckland, New Zealand; [11]Max-Planck-Institute for Ornithology, Seewiesen, Germany; [12]Department of Anthropology, University of Zurich, Zurich, Switzerland; [13]Animal Ecology Group, Department of Biology, Utrecht University, Utrecht, Netherlands

*For correspondence:
lisa.horn@univie.ac.at

Competing interests: The authors declare that no competing interests exist.

**Abstract** The investigation of prosocial behavior is of particular interest from an evolutionary perspective. Comparisons of prosociality across non-human animal species have, however, so far largely focused on primates, and their interpretation is hampered by the diversity of paradigms and procedures used. Here, we present the first systematic comparison of prosocial behavior across multiple species in a taxonomic group outside the primate order, namely the bird family Corvidae. We measured prosociality in eight corvid species, which vary in the expression of cooperative breeding and colonial nesting. We show that cooperative breeding is positively associated with prosocial behavior across species. Also, colonial nesting is associated with a stronger propensity for prosocial behavior, but only in males. The combined results of our study strongly suggest that both cooperative breeding and colonial nesting, which may both rely on heightened social tolerance at the nest, are likely evolutionary pathways to prosocial behavior in corvids.

## Introduction

The investigation of prosocial behavior (i.e. voluntary actions that benefit another individual at no or low costs to the actor; *Marshall-Pescini et al., 2016*), is of particular interest from an evolutionary point of view, because the act of benefitting another individual without receiving a direct gain to oneself represents an evolutionary puzzle (*Clutton-Brock, 2009*; *Riehl, 2013*). Humans show high levels of prosocial behaviors from an early age on (*Silk and House, 2011*), although their expression and developmental trajectories are subject to cross-cultural and societal variation (*House et al., 2020*). The importance of prosociality for human interactions has inspired comparative studies on

the evolutionary origin of this trait. The majority of experimental studies in non-human animals have focused on primates (for a review, see *Marshall-Pescini et al., 2016*), but recent research revealed prosocial tendencies also in other mammals (e.g. domestic dogs [*Quervel-Chaumette et al., 2016*]; wolves [*Dale et al., 2019*]; rats [*Ben-Ami Bartal et al., 2011*; *Schweinfurth and Taborsky, 2018a*]) and several bird species (e.g. azure-winged magpies [*Horn et al., 2016*; *Massen et al., 2020*]; pinyon jays [*Duque et al., 2018*]; African grey parrots [*Brucks and von Bayern, 2020*]). Nevertheless, not all tested species have shown prosocial tendencies (e.g. chimpanzees [*Silk et al., 2005*]; cottontop tamarins [*Cronin et al., 2009*]; meerkats [*Amici et al., 2017*]; common ravens [*Di Lascio et al., 2013*; *Lambert et al., 2017*; *Massen et al., 2015a*]). Following these variable initial results, the importance of understanding which social factors and which characteristics of a species' social system may underlie the expression of prosociality across non-human animal species became particularly evident. Unfortunately, however, comparisons of prosociality across species have been hampered by the diversity of paradigms and procedures used (*Marshall-Pescini et al., 2016*).

The most comprehensive experimental investigation of prosocial behavior in primates tested 15 species (including human children) in the same experimental set up, that is the group service paradigm (hereafter GSP; *Burkart et al., 2014*). In the GSP, individuals are tested in their regular social group and can make food available to other group members by operating a simple mechanism, without obtaining any food themselves. *Burkart et al., 2014* showed that species-specific prosocial tendencies in the GSP were best explained by the degree of allomaternal care (i.e. offspring care by individuals other than the mother) across the tested species. These results were in line with the cooperative breeding hypothesis, which states that 'cooperative breeding is accompanied by psychological changes leading to greater prosociality' (*Burkart et al., 2009*). Additional factors positively influencing the amount of prosocial behavior – albeit to a lesser degree than allomaternal care – were the presence of monogamous pair bonds and high social tolerance (i.e. equal access to food for all group members) measured during the GSP (*Burkart et al., 2014*). The latter result fits the self-domestication hypothesis (*Hare et al., 2012*; *Hare, 2017*), according to which prosociality arises as a by-product of selection against reactive aggression – particularly in males (*Wrangham, 2019*) – and selection for increased tolerance (see *Sánchez-Villagra and van Schaik, 2019* for a critical appraisal of historical and current theories on self-domestication). While both the cooperative breeding hypothesis and the self-domestication hypothesis acknowledge an underlying link between increased social tolerance and prosociality, the cooperative breeding hypothesis puts emphasis on allomaternal offspring care, whereas the self-domestication hypothesis suggests the decrease of reactive aggression as the crucial factor for the emergence of human-like prosociality. The comparative approach is particularly promising for distinguishing between these hypotheses (*Burkart et al., 2014*). However, concentrating solely on the primate order offers only one perspective on the evolution of prosocial behavior, which has also been criticized because of possible effects of common ancestry (e.g. cooperative breeding in primates occurs only in two taxonomic groups – humans and callitrichid monkeys; *Thornton and McAuliffe, 2015*). Hence, applying a standardized comparative approach to other taxonomic groups would be paramount for drawing more general conclusions (*Beran et al., 2014*).

From a comparative perspective, the corvid family, which is a cosmopolitan bird taxon that includes crows, ravens, jays, and magpies, is of particular interest for the investigation of prosociality. Corvids have similar neuron counts compared to many primate species (*Olkowicz et al., 2016*) and show similarly complex cognitive traits (*Taylor, 2014*; *Güntürkün and Bugnyar, 2016*; *Boucherie et al., 2019*). Most corvid species are long-lived, highly social (*Emery et al., 2007*) and pair bonds are extremely strong, even lifelong in some species (*Henderson et al., 2000*). About 40% of all extant corvid species from several separate genera are cooperative breeders (defined in birds as more than the two parents caring for the brood; for example azure-winged magpies, carrion crows; see *Cockburn, 2006*; *Griesser et al., 2017*). Since related as well as unrelated helpers have been documented to contribute to offspring care, both kin selection (*Green et al., 2016*) and pay-to-stay strategies (*Kingma, 2017*) seem important to explain cooperative breeding in birds. Additionally, a number of corvids breed colonially, where several pairs nest in physical proximity, including rooks, Eurasian jackdaws, and azure-winged magpies (see *Madge and Burn, 1999*). It has been argued that relaxed territorial defense, reduced reactive aggression, and increased tolerance toward conspecifics may lead to the emergence of colonial nesting in birds (*Brown, 1974*). Consequently, corvids' variation in cooperative breeding and colonial nesting make them the optimal candidates

for testing both the cooperative breeding hypothesis and the self-domestication hypothesis in a lineage other than primates.

Previous experiments in corvids have demonstrated prosocial behavior in azure-winged magpies (using the GSP *Horn et al., 2016* as well as an active food-sharing paradigm *Massen et al., 2020*) and pinyon jays (using a prosocial choice task *Duque et al., 2018*). Both species breed cooperatively (*Cockburn, 2006*) and nest in colonies, with several breeding pairs nesting in close proximity (*Madge and Burn, 1999*). Additionally, there has been tentative evidence for prosocial tendencies in a prosocial choice task in Eurasian jackdaws (*Schwab et al., 2012*), which also nest in colonies, but do not breed cooperatively (*Cockburn, 2006*). In contrast, subadult ravens for example, which are able to cooperate with a conspecific partner to receive mutual rewards (*Massen et al., 2015b*; *Asakawa-Haas et al., 2016*), have so far not shown any evidence of prosociality, despite having been tested with multiple experimental paradigms (e.g. different prosocial choice tasks [*Di Lascio et al., 2013*; *Lambert et al., 2017*]; token exchange task [*Massen et al., 2015a*]). While ravens tend to form groups for foraging and roosting as non-breeders (*Heinrich, 1989*; *Loretto et al., 2016*), they are highly territorial during breeding (*Boucherie et al., 2019*) and it is not clear whether the characteristics of their social system contribute to their apparent lack of prosocial tendencies. To disentangle the influence of cooperative breeding and colonial/territorial nesting, respectively, on prosociality, it is necessary to test a sample of different species that vary along these factors, and to avoid differences that result from methodological heterogeneity by using the same standardized procedure.

Here, we present the first systematic comparison of prosocial behavior across multiple species in a taxonomic group outside the primate order. We measured prosociality in 11 social groups of eight corvid species (total N = 72 individuals), which were all highly social (i.e. living and foraging in social groups during at least some stages of their life history; *Komeda et al., 1987*; *Uhl et al., 2019*; *Kubitza et al., 2015*; *Clayton and Emery, 2007*; *Braun et al., 2012*; *Miyazawa et al., 2020*; *Holzhaider et al., 2011*; *Ekman and Griesser, 2016*), but varied in the expression of cooperative breeding and colonial nesting (*Figure 1d*). We used a standardized experimental paradigm developed in primates (i.e. the GSP; *Burkart et al., 2014*), which has recently been adapted and successfully applied in birds (*Horn et al., 2016*). To keep the results comparable, we kept the procedures as similar as possible to the original study with primates (*Burkart et al., 2014*). In the prosocial test of the GSP, individuals can land on the provisioning perch of the apparatus, and consequently make food available to their group members via a seesaw mechanism (*Figure 1a*). Crucially, the bird on the provisioning perch cannot obtain any food itself and it has to remain on the provisioning perch until another individual arrives on the other side of the apparatus (position 1; see *Figure 1b*) to take the food (see *Video 1*; see Materials and methods section for details). Habituation, training and two control conditions (i.e. empty control: no food available; blocked control: access to food blocked; see *Video 2* and *Video 3*) ascertain that the individuals understand the experimental task and that landing on the provisioning perch in the prosocial test does not reflect the absence of sufficient inhibitory control (*Figure 1c*). In addition to prosocial tendencies, the GSP also measures how even the access to sequentially provided food is across the individuals of a given social group (i.e. whether one or few individuals monopolize the food source and obtain most of the food or whether similar numbers of food pieces are obtained by all group members; *Figure 1c*). In primates, this evenness score has been used as a proxy for social tolerance (*Burkart et al., 2014*).

To assess the explanatory value of cooperative breeding and colonial nesting for prosocial behavior in corvids, we used linear regression models and an information-theoretic approach to model selection and model averaging. Additionally, since sex differences have been observed in prosocial food sharing in natural observations (*von Bayern et al., 2007*; *Scheid et al., 2008*; *Chiarati et al., 2011*) and experiments (*Schwab et al., 2012*), we also included the individuals' sex into the model. Further, to test the extent to which common ancestry affected the birds' prosocial tendencies, we calculated a phylogenetically controlled mixed-effects model (for phylogenetic relationships between the tested species, see *Figure 1—figure supplement 1*). Finally, because within a species prosocial behavior might be expressed differently between the sexes (*Massen et al., 2020*; *Schwab et al., 2012*; *von Bayern et al., 2007*) and between age classes (*Chiarati et al., 2011*), we also examined intraspecific provisioning patterns.

Our results demonstrate that cooperative breeding is positively associated with the expression of prosocial behavior in corvids, although this effect is qualified by interactions between sex and both

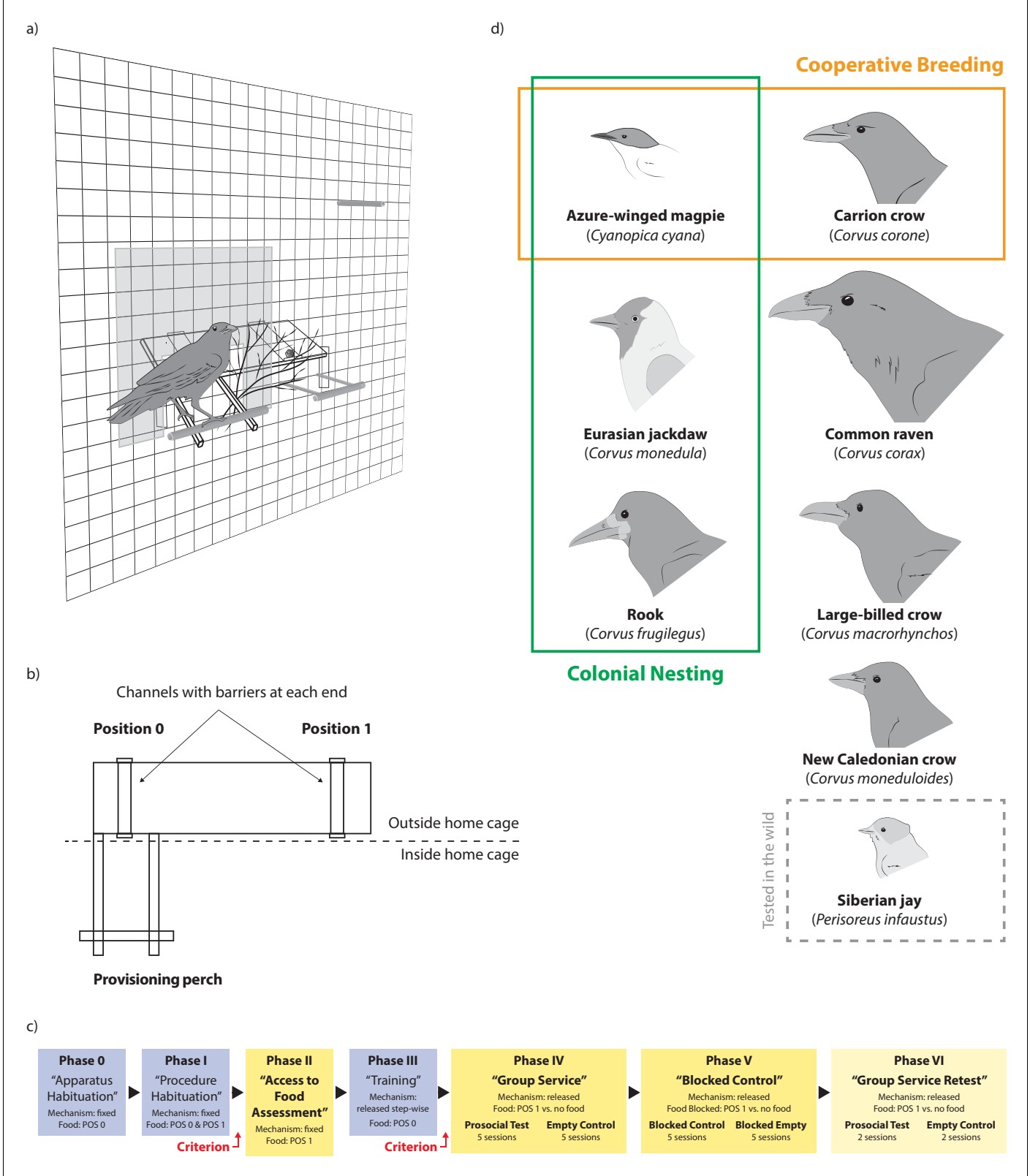

**Figure 1.** Overview of the study design and set-up. (**a**) Experimental set-up as seen from the inside of the aviary with a bird sitting on the provisioning perch, thereby making food available to the group. (**b**) Schematic of the apparatus with location of positions 0 and 1 in relation to the provisioning perch. (**c**) Experimental procedure; habituation and training phases are given in blue, test phases are given in yellow; subjects needed to reach a given criterion to be included in the analysis of phases II and IV-VI; see supplementary information for details. (**d**) Overview of the tested species and their key

*Figure 1 continued on next page*

*Figure 1 continued*

social system differences; orange boxes represent the presence of obligate or facultative cooperative breeding for the respective species, green boxes represent the presence of colonial nesting.

The online version of this article includes the following figure supplement(s) for figure 1:

**Figure supplement 1.** Phylogenetic tree of the tested species.

the factors cooperative breeding and colonial nesting, which were also important for explaining the occurrence of prosocial behavior in the birds. Additional separate analyses for the two sexes showed that both cooperative breeding and colonial nesting positively affected prosociality, albeit differently for the two sexes. While the effect of cooperative breeding seemed to be driven by females' prosociality, colonial nesting only predicted males' prosocial actions. The phylogenetically controlled model confirmed the importance of both cooperative breeding and colonial nesting and showed that the phylogenetic signal was weak in terms of prosocial behaviors in corvids. Same-sex provisioning dyads were equally common as opposite-sex dyads and we observed both provisioning from adults to juveniles and vice versa. Our results highlight that both alloparental care and increased social tolerance are important evolutionary trajectories for the emergence of prosocial behavior in birds.

## Results

### Between-species variation in prosocial provisioning and evenness of access to food

Across all species and groups, the amount of food provided by those birds that discriminated between the prosocial test and both control conditions (i.e. landed significantly more often on the provisioning perch when they could provide food to their group members than when there was no food or when access to the food was blocked for the recipient; N = 12; four azure-winged magpies, two carrion crows, two Eurasian jackdaws, one rook, one New-Caledonian crow, one common raven, one large-billed crow; see *Appendix 1—table 1*), showed high variability and ranged from 0% to 98% (*Table 1*). The evenness of the birds' access to food within the group, which was measured in a different phase of the experiment (see Appendix 2) and which has been proposed as a proxy for social tolerance in primates in the original study (*Burkart et al., 2014*), was medium to high in all tested species (cf. 20; *Table 1*) and was not correlated with provided food values across groups (Spearman's rho = −0.326, p=0.327, N = 11).

### Linking cooperative breeding and colonial nesting with prosocial behavior

The averaged model identified the main factors sex and cooperative breeding as having a high explanatory degree for the number of landings on the provisioning perch in the prosocial test (i.e. making food available for conspecifics; see *Figure 1a* and *Video 1*; model results in *Figure 2—source data 1*). Overall, individuals from cooperatively breeding species landed more often on the provisioning perch than individuals from non-cooperatively breeding species (*Figure 2a*), and males landed more often than females (*Figure 2b*). These main effects were qualified by the high explanatory degree of the interaction terms of both cooperative breeding and nesting type with sex (*Figure 2—source data 1*), meaning that the main effects were conditional upon one another.

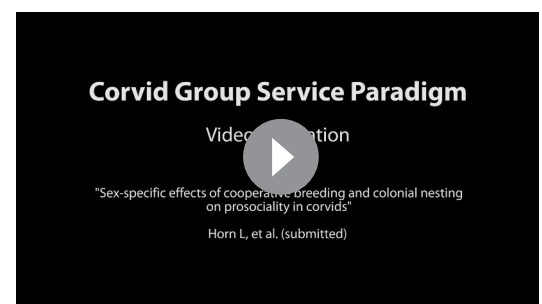

**Video 1.** Prosocial test. Example videos of prosocial test trials taken from three species (i.e. azure-winged magpies, carrion crows, common ravens). Food is placed on the recipient side (position 1). Food can be provided to a group member, if an individual lands on the provisioning perch.
https://elifesciences.org/articles/58139#video1

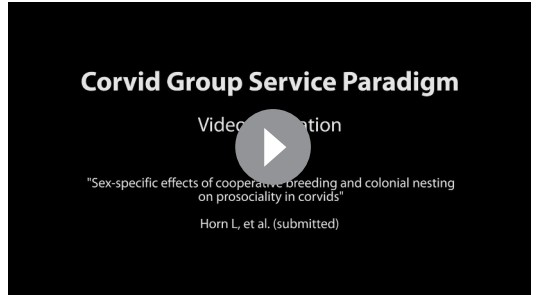

**Video 2.** Empty control. Example videos of empty control trials taken from three species (i.e. azure-winged magpies, carrion crows, common ravens). No food is placed on the recipient side (position 1). Therefore, no food can be provided to group members.
https://elifesciences.org/articles/58139#video2

**Video 3.** Blocked control. Example videos of blocked control trials taken from three species (i.e. azure-winged magpies, carrion crows, common ravens). Food is placed on the recipient side (position 1), but access to the food is blocked with a fine net. Therefore, although food is visible, no food can be provided to the group members.
https://elifesciences.org/articles/58139#video3

In order to ascertain the robustness of our model, we re-did the analysis, always excluding one species at a time. Four out of eight models had the same results as before (removed species: Siberian jays, N = 48; rooks, N = 48; common ravens, N = 44; carrion crows, N = 45), while nesting type had an added high explanatory degree in two models (removed species: New-Caledonian crows, N = 46; azure-winged magpies, N = 43). In one model nesting type, sex, and the interaction between these two factors had a high explanatory degree, while cooperative breeding and the interaction between cooperative breeding and sex were only marginally important (i.e. $SW_{AICc}$ = 0.44; removed species: large-billed crows, N = 42). Finally, in one model the intercept-only model was included in the selection of best-fitting models (removed species: Eurasian jackdaws, N = 41),

**Table 1.** Prosocial food provisioning and evenness of access to food across all tested species and groups.
Given are the classifications of cooperative breeding and nesting type for the tested species, as well as the percentage of food provided in the prosocial test and Pielou's J' as a measure for evenness of access to food for each of the groups.

| Species | Cooperative breeding[*] | Nesting type[†] | Group (N) | Phase IV provided food[‡] | Phase II Pielou's J' |
|---|---|---|---|---|---|
| Azure-winged magpie | Yes | Colonial | 1 (5) | 98% | 0.72 |
| | | | 2 (4) | 64% | 0.83 |
| Carrion crow | yes | Territorial | 1 (6) | 57% | 0.46 |
| Eurasian jackdaw | no | Colonial | 1 (14) | 33% | 0.73 |
| Rook | no | Colonial | 1 (12) | 2% | 0.86 |
| New-Caledonian crow | no[§] | Territorial | 1 (3) | 70% | 0.52 |
| | | | 2 (2) | 0% | 0.36 |
| Common raven | no | Territorial | 1 (9) | 21% | 0.73 |
| Large-billed crow | no | Territorial | 1 (9) | 16% | 0.97 |
| Siberian jay | no | Territorial | 1 (5) | 0% | 0.82 |
| | | | 2 (3) | 0% | 0.91 |

[*]Classifications after (**Cockburn, 2006**).
[†]Classifications after (**Madge and Burn, 1999**).
[‡]In line with the original publication (**Burkart et al., 2014**), provided food was calculated as the corrected percentage of food provisioning per group in the last two test sessions of the prosocial test, only by those individuals that passed the criterion of landing significantly more often in the test compared to both control conditions. Note that raw and corrected measures of food provisioning are highly correlated (Spearman's rho = 0.892, p≤0.001, N = 11).
[§]Occurrence of cooperative breeding is classified as unknown, but assumed as absent according to **Cockburn, 2006**.

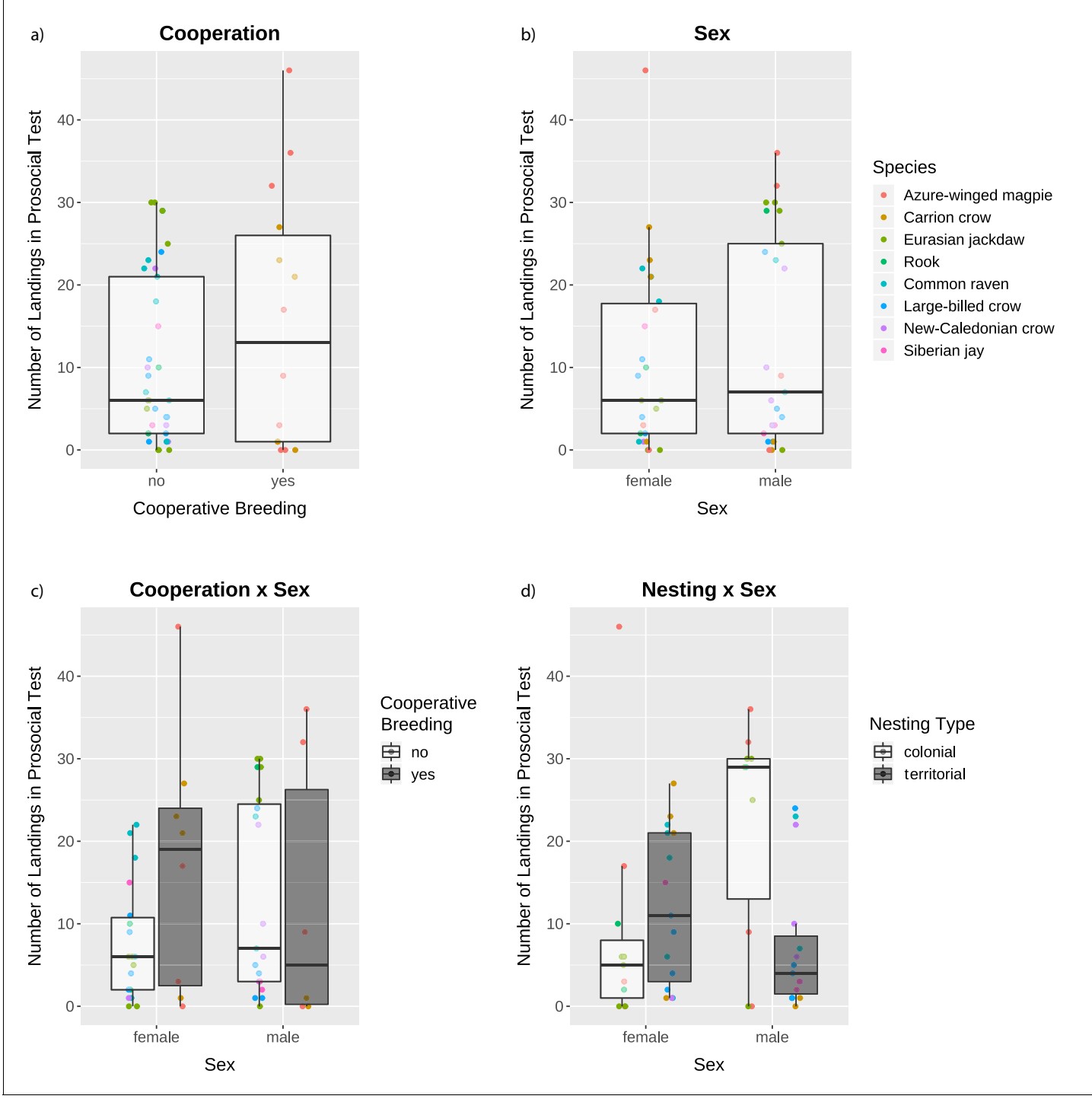

**Figure 2.** Number of landings in the prosocial test as a function of the factors with a high explanatory degree. The box plots represent medians (horizontal lines), inter-quartile ranges (boxes), as well as minima and maxima (whiskers). All data are represented with dots. Dots not encompassed by the whiskers are outliers. Dot colors in all panels indicate the species according to the legend in the top right panel.

The online version of this article includes the following source data for figure 2:

**Source data 1.** Effects of cooperative breeding, nesting type, and sex on the number of landings in the prosocial test.

**Source data 2.** Effects of cooperative breeding and nesting type on the number of landings in the prosocial test in female birds (**A**) and male birds (**B**).

implying that the averaged model was not robust. Overall, these results are consistent and corroborate the robustness of our original results. We specifically note that the Siberian jays were the only species tested in the wild and that they did not successfully provide food to their group members, which could have been an artifact of them being tested in the wild rather than their social system. The fact that the results remained practically identical after excluding the Siberian jays (see *Appendix 1—table 4*) suggests that the results obtained with the complete dataset were not driven by the Siberian jays per se.

When splitting the data by sex due to the high explanatory degree of the interaction terms, our analyses showed that for males (N = 25) the factor colonial nesting had a high explanatory degree (Estimate = −15.066, SE = 4.528, z = 3.154, $SW_{AICc}$ = 1.00, $N_{Models}$ = 2): males from colonial species landed more often than males from territorial species (*Figure 2d*). Cooperative breeding had only a very low explanatory degree in males (*Figure 2c*; see *Figure 2—source data 2* for full model results). In contrast, for the females (N = 26) the factor cooperative breeding had a high explanatory degree (Estimate = 9.686, SE = 4.427, z = 2.076, $SW_{AICc}$ = 1.00, $N_{Models}$ = 2): females from cooperatively breeding species landed more often than females from non-cooperatively breeding (*Figure 2c*). Nesting type had only a very low explanatory degree in females (*Figure 2d*; see *Figure 2—source data 2* for full model results). Using the same procedure of excluding one species at a time as above, we could ascertain the robustness of the model including only the males: all eight models had the same results as before (see Appendix 1 for details). Additionally, the male birds from colonial species landed significantly more often on the provisioning perch than the male birds from territorial species, when only testing for the factor nesting type (Welch t-test: t = 3.01, df = 13.66, p-value=0.005). The model including only the females, however, was less robust: only two out of eight models had the same results as before, while in five models the intercept-only model was included in the selection of best-fitting models (see Appendix 1 for details). Also when testing only whether the females from cooperatively breeding species landed more often on the provisioning perch than the females from species that do not breed cooperatively, the results were only marginally significant (Welch t-test: t = −1.64, df = 8.30, p-value=0.069).

When looking only at the landings of the birds that discriminated between the prosocial test and both control conditions (N = 12), we found that there was a non-significant trend for the birds from colonial species to land more often on the provisioning perch (N = 7, median = 30, IQR = 29–34) than the birds from territorial species (N = 5, median = 23, IQR = 22–24; Mann-Whitney: W = 30, p=0.0505). The birds from cooperatively breeding species (N = 6, median = 29.5, IQR = 24–35) did not differ significantly in the number of their landings from the individuals from species that do not breed cooperatively (N = 6, median = 26.5, IQR = 22.5–29; Mann-Whitney: W = 13, p=0.470).

## Testing the effect of phylogeny on prosocial behavior

As in the original model, also a phylogenetically controlled model showed that the main factors cooperative breeding and sex significantly predicted the number of landings on the provisioning perch in the prosocial test (cooperative breeding: estimate = 10.001, 95% HPD interval [0.082, 19.886], $P_{mcmc}$ = 0.048; sex: estimate = 19.660, 95% HPD interval [8.899, 30.292], $P_{mcmc}$ = 0.0002), and that these main effects were again qualified by significant interactions between both cooperative breeding and sex (Estimate = −16.394, 95% HPD interval [−30.183,−2.329], $P_{mcmc}$ = 0.020) and nesting type and sex (Estimate = −20.576, 95% HPD interval [−33.588,−8.551], $P_{mcmc}$ = 0.002; see *Appendix 1—table 5* for full model results). The phylogenetic signal was weak (mean λ = 0.035; posterior mode = 0.001; 95% HPD interval [0.000, 0.185]).

## Dyad-level variation in prosocial provisioning

Opposite-sex provisioning did not occur more often than same-sex provisioning in the tested species, both when considering all individuals in each group (Wilcoxon: N = 7, T+=7, p=0.271) and when only considering provisioning by these individuals that discriminated between the test and the control conditions in each group (N = 7, T+=10, p=0.553). There were species differences in the distribution of sex dyad types, which could, however, not be linked back to either cooperative breeding or nesting type (see *Figure 3* and *Figure 3—source data 1* for details). With regard to age-dependent provisioning, we had very little data, as only five groups from three species contained juvenile individuals (i.e. azure-winged magpie group 2; New Caledonian crow groups 1 and 2; Siberian jay

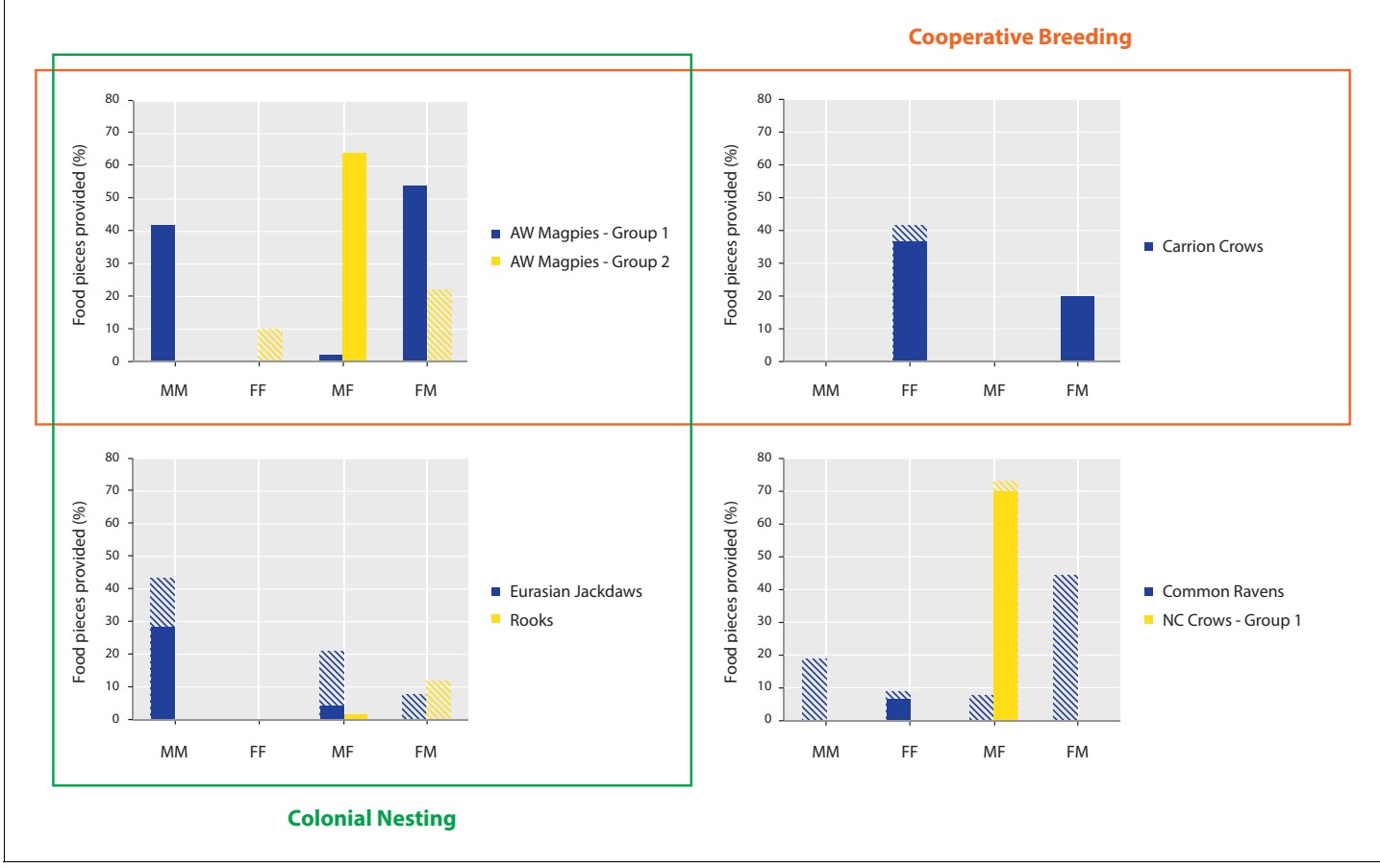

**Figure 3.** Distribution of food provisioning per dyad sex composition. The bars represent the percentage of food provided in the last two test sessions of the prosocial test in those seven groups where provisioning occurred and for which we had data on the individuals' sex and the dyad identities. Full bars comprise the individuals that passed the criterion of landing significantly more in the test versus both control conditions. Striped bars comprise all individuals. Dyad types: male donor – male recipient (MM), female donor – female recipient (FF), male donor – female recipient (MF), female donor – male recipient (FM). All possible dyads: azure-winged magpies, group 1, 3 MM, 1FF, 6MF/FM; azure-winged magpies, group 2, 0 MM, 3FF, 3MF/FM; carrion crows, 1 MM, 6FF, 8MF/FM; Eurasian jackdaws, 21 MM, 21FF, 49MF/FM; rooks 15 MM, 15FF, 36MF/FM; common ravens, 6 MM, 10FF, 20MF/FM; New Caledonian crows, group 1, 1 MM, 0FF, 2MF/FM.

The online version of this article includes the following source data for figure 3:

**Source data 1.** Data on food provisioning per dyad sex composition.

groups 1 and 2). While there was no uniform pattern among those four groups, we did witness a juvenile providing food to adults in the azure-winged magpie group (33% of total provided food) and the one New Caledonian crow group where provisioning occurred (95% of total provided food).

## Discussion

Our results reveal that cooperative breeding is positively associated with the propensity for prosocial behaviors in corvids, but that this main effect is qualified by an interaction with sex. Additional separate analyses for the two sexes showed that both cooperative breeding and colonial nesting positively affected prosociality, albeit differently for the two sexes. Consequently, our results support both the cooperative breeding hypothesis (*Burkart et al., 2014*; *Burkart et al., 2009*), which emphasizes the role of allomaternal care, and the self-domestication hypothesis (*Hare et al., 2012*; *Hare, 2017*; *Wrangham, 2019*), which stresses the importance of low levels of reactive aggression and high levels of social tolerance, as explanatory approaches for the evolution of prosocial behavior in the corvid family. An additional model that controlled for common ancestry confirmed the

importance of both cooperative breeding and colonial nesting and showed that the phylogenetic signal was weak in terms of prosocial behaviors in corvids.

The conclusion that both cooperative breeding and colonial nesting positively affect prosocial behavior in corvids is corroborated by the species-specific provisioning rates in our study: provisioning was particularly high in the cooperatively breeding, colonially nesting azure-winged magpies (64–98%), high in the facultative cooperatively breeding, territorially nesting carrion crows (57%), and intermediate in the non-cooperatively breeding, colonially nesting jackdaws (33%). In the third colonial species, the rooks, very little food was provided during the prosocial test (2%). However, in this group only three out of 12 individuals could be habituated to the apparatus despite extensive training (see *Appendix 1—table 1*). Therefore, it is possible that the limited number of possible donors and recipients prevented higher provisioning rates. The provisioning results obtained in jackdaws parallel previous findings with this species when tested in a dyadic prosocial choice paradigm: in that study the jackdaws also provided food for their conspecifics in certain contexts (e.g. more provisioning for opposite-sex recipients; *Schwab et al., 2012*). Carrion crows, however, were previously not found to exhibit prosocial tendencies in a token exchange paradigm where they had the opportunity to transfer tokens to a conspecific partner, which could in turn be exchanged for food (*Wascher et al., 2020*). The authors of that study argued that the exchange paradigm might have been too complicated for the birds, as it required understanding the value of the tokens (*Wascher et al., 2020*). It is therefore possible that the GSP used in the current study, which simply required the birds to land on the provisioning perch in order to make food available for the group members, made it easier for the carrion crows to express their prosocial behavior.

An interesting exception to the predicted pattern is the remarkably high provisioning rate in one group of New Caledonian crows (70%). According to *Cockburn, 2006*, however, it remains unknown whether this species might engage in cooperative breeding. New Caledonian crows allow their offspring to stay in the parental territory for up to two consecutive breeding seasons (*Holzhaider et al., 2011*) and feed the juveniles for up to ten months post-fledging (*Hunt et al., 2012*). Helping by offspring at the nest has never been documented (*Holzhaider et al., 2011*), but it has proven difficult to observe interactions at the nest in most habitats. Family living, where offspring delay dispersal from the parental territory beyond nutritional independence, has been suggested as one of the evolutionary routes to cooperative breeding (*Griesser et al., 2017*; *Brown, 1974*). Nevertheless, Siberian jays, which also live in family groups (*Griesser et al., 2017*), showed no prosocial behavior in our study. However, one has to consider, that the Siberian jays were the only species not tested in captivity but in the wild. Although the study population is well habituated to the presence of humans and to field experiments (*Griesser, 2013*), it is possible that they were not as focused on the experiment as the captive species. Additionally, Siberian jays fully rely on scatter hoarding to survive the winters at the study site. Since this behavior has been shown to be predominantly selfish (*Ekman et al., 1996*), it is possible that Siberian jays' particular feeding strategy explains their lack of prosocial tendencies. Therefore, to advance understanding of the role of family living in regard to prosociality, it would be important to further investigate prosocial tendencies in other family-living corvid species (*Uomini et al., 2020*).

In line with the original comparative study using the GSP in primates (*Burkart et al., 2014*), we corrected the percentage of provided food by including only provisioning by individuals that passed the criterion of landing significantly more often in the test compared to both control conditions, thereby giving the most conservative measure of prosocial provisioning. The rate of individuals that passed this criterion ranged from 0% to 50% across species (see *Appendix 1—table 1* for details). Due to the experimental paradigm (i.e. the group setting) it is difficult to unequivocally conclude that the individuals that did not pass this criterion did not understand the task. These individuals might have been willing to land on the apparatus in the prosocial test, but might have simply been slower than other group members. Alternatively, they might have understood the difference between the prosocial test and the two control tasks, but they might just not have had prosocial tendencies. Given that explanation, it is interesting to note that the percentage of birds that passed the criterion was relatively high in the cooperatively breeding and/or colonial species (azure-winged magpies: 4 out of 8 birds (50%); carrion crows: 2 out of 6 (33%); Eurasian jackdaws: 2 out of 10 (20%); rooks: 1 out of 3 (33%)) and relatively lower in most territorial species (New Caledonian crows: 1 out of 5 (20%); common ravens: 1 out of 7 (14%); large-billed crows: 1 out of 9 (11%); Siberian jays: 0 out of 7 (0%)), especially as there is no reason to assume that the latter are cognitively less

developed than the former (see e.g., *Güntürkün and Bugnyar, 2016*). When looking at the number of landings on the provisioning perch (i.e. with and without actual provisioning) of these 12 birds that passed the criterion, we only found a non-significant trend of more landings from the birds from colonial species than from the birds from territorial species and no effect of whether the birds came from a cooperatively breeding species or not. However, an individual would only pass this criterion if it landed relatively often on the provisioning perch in the prosocial test, thereby inherently also demonstrating a prosocial tendency (*Burkart et al., 2014*). Therefore, it is not surprising that among this group of birds, no strong differences according to cooperative breeding and colonial nesting became apparent with regard to prosocial tendencies. Additionally, the small sample size did not allow us to include further factors in the analysis (e.g. sex) and might have hampered the detection of potential differences.

Across all individuals we found that sex modulated the effects of both cooperative breeding and colonial nesting on how often the birds landed on the provisioning perch. The positive effect of cooperative breeding on the number of landings in the prosocial test was mainly driven by the females, although the results of the female-only model were not very robust. Nevertheless, the fact that the females from cooperatively breeding species were particularly prosocial is surprising, because observations in wild populations showed that in many cooperatively breeding corvids only a minority of the helpers were females (e.g. azure-winged magpies [*Ren et al., 2016*], their closely related sister species the Iberian magpies [*Valencia et al., 2003*], carrion crows [*Baglione et al., 2002*]) and that male helpers provided more care during breeding than females (e.g. carrion crows [*Canestrari et al., 2005*]). However, due to the high energetic demand of incubation that usually only the females incur, cooperatively breeding females might depend more on helpers' contributions than males and they might use acts of prosocial behavior throughout the year (note that our studies were all conducted outside the breeding season) to incentivize group members to remain in the group. This argument is in line with the interdependence hypothesis, which states that cooperative acts are expected most when individuals strongly rely on each other (*Roberts, 2005*). In contrast, among colonially nesting birds, male individuals, but not females, were particularly prosocial, together with an overall main effect of stronger prosocial tendencies in males than in females across all tested groups. According to costly signaling theory (*Zahavi, 1997*), prosocial acts can be regarded as honest signals that advertise the donor's underlying qualities (e.g. health, strength, ability to control resources; cf. competitive altruism hypothesis; *Hardy and Van Vugt, 2006*). Based on these premises, dominant individuals would be expected to show more prosocial behavior than subordinates. This prediction has been supported by experimental evidence from birds (*Duque and Stevens, 2016*) and several primate species (e.g. long-tailed macaques; *Massen et al., 2010*; for a review, see *Marshall-Pescini et al., 2016*). In most corvids, males are dominant over females (*Scheid et al., 2008*; *Massen et al., 2014*; *Wechsler, 1988*; *Ode et al., 2015*; *Chiarati et al., 2010*; *Sklepkovych, 1997*) and would therefore be expected to face greater pressure to advertise their dominance rank than females. This might be most evident in colonial males, which nest in close proximity with many conspecifics and consequently engaging in dominance challenges might be particularly costly for them (*Verhulst and Salomons, 2004*). The self-domestication hypothesis also emphasizes the importance of reduced reactive aggression and violent conflict between male individuals, not females, as an important factor for the evolution of human-like prosociality (*Wrangham, 2019*). Beyond that, prosocial actions might represent an attempt of males to trade food for extra-pair copulations (*Tryjanowski and Hromada, 2005*) or to maintain relationships with affiliative partners other than the mated partner (*Miyazawa et al., 2020*; *von Bayern et al., 2007*; *Boucherie et al., 2016*). In general, the results from the single sex models – especially the female-only model – have to be considered preliminary due to the low sample size. Future studies with larger sample sizes and experiments that specifically address these sex differences are needed to reveal which of these hypotheses explain the sex-specific effects of both cooperative and colonial breeding in birds.

One additional limitation of this study was that, despite the considerable research effort of this multi-lab study, we only managed to test few replicates per species. Finding test populations is a common problem for large-scale comparative studies (e.g. the original GSP study in primates [*Burkart et al., 2014*]; see also *Morales Picard et al., 2020*; *O'Hara et al., 2017*; *MacLean et al., 2014*; *Many Primates et al., 2019*). While the provisioning rates were similar in the two groups of azure-winged magpies (i.e. the two highest provisioning rates at 98% and 64%, respectively) and the

two groups of Siberian jays (i.e. both 0%), there was a substantial difference between the two groups of New-Caledonian crows: in one group, 70% of the available food was provided to the group members, whereas in the other group there was no successful provisioning at all. One has to consider, though, that the latter group consisted only of two individuals at the time of testing. Therefore, similarly as in the rooks, the limited number of potential donors and recipients might have prevented successful provisioning. Also, within the groups, there was obvious inter-individual variation, with some individuals providing the majority of the food to their group members while other individuals rarely landed on the provisioning perch at all. Due to the unrestricted group setting of the GSP, it is not possible to discern whether the individuals that did not land on the provisioning perch were not motivated to provide food for their group members, or whether they were merely too slow to do so compared to other individuals that for example were bolder or faster. To be able to more confidently demonstrate true species generalizations and rule out a strong effect of individual characteristics, future studies should attempt to increase the number of replicates per species and should bolster the results of the GSP with individual testing paradigms (e.g. prosocial choice experiments).

Within the groups, we would have expected more opposite-sex provisioning than same-sex provisioning, especially from males to females. Observations of naturally occurring food sharing suggest that food provisioning in corvids might serve the function of forming pair bonds and social bonds in general (*Massen et al., 2015a*; *Miyazawa et al., 2020*; *von Bayern et al., 2007*). Also in the context of a prosocial choice experiment with jackdaws, opposite-sex recipients were more likely to elicit prosocial behavior from the donors than same-sex recipients (*Schwab et al., 2012*). Similarly, in an active food-sharing paradigm, azure-winged magpies shared high-value food items preferably with, although not restricted to, members of the opposite sex (*Massen et al., 2020*). However, both opposite-sex and same-sex provisioning occurred equally often in our study. This might have been because of the constraints of the GSP, where food is made available to the whole group and the donor has limited possibilities to influence the specific recipient of its prosocial action. There were some differences in the distribution of donor-recipient sex-constellations, which could, however, not be linked back to either cooperative breeding or nesting type, but were more likely a result of the specific group compositions. Interestingly, although the majority of the tested birds were adults, many instances of juveniles providing food to adults were observed in both the azure-winged magpies and the New Caledonian crows, which accounted for almost all the provided food in the latter species. Prosocial acts from juveniles are expected in cooperatively breeding species based on observations in the wild (e.g. Iberian magpies [*Valencia et al., 2003*]; carrion crows [*Baglione et al., 2002*]). In contrast, New Caledonian crow parents feed their juvenile offspring for extended periods (*Hunt et al., 2012*), while food provisioning by juveniles has never been documented (*Holzhaider et al., 2011*). Our finding of prosocial behavior in a juvenile New Caledonian crow underlines the importance of considering the role of family living in the absence of cooperative breeding for the evolution of prosociality in birds (*Uomini et al., 2020*). Future studies, where samples show larger age variation within the groups or where the same groups can be tested at different time points with differing age ratios, would also be very informative regarding the question of the influence of age on prosocial behavior (*Kaplan, 2020*). An additional factor that has been argued to play an important role for prosocial acts between individuals is their relatedness (e.g. *Bourke, 2014*; but see *Schweinfurth and Taborsky, 2018b*). However, since kinship relations between individuals were unknown for about half of the groups tested in the current study, we were not able to include kinship as a factor in the analysis. Future studies that track relatedness between group members could further investigate the relevance of this factor for prosocial behavior in corvids.

In contrast to the comparative study on primate prosociality (*Burkart et al., 2014*), we did not use the degree of allomaternal care, but rather a nominal classification as either cooperatively breeding species or not as a predictor in our models. The reasoning for that change was two-fold. First, there is less information on the specific number and degree of investment of helpers in cooperatively breeding bird species (*Cockburn, 2006*) compared to primates (*Isler and van Schaik, 2012*) and even within the same corvid species, the numbers seem to differ greatly depending on the population (e.g. *Komeda et al., 1987*; *Ren et al., 2016*; *Valencia et al., 2003*; *Baglione et al., 2002*). Second, since we only had two cooperatively breeding species in our sample, a more detailed representation of the degree of cooperative breeding would have decreased the statistical power of our analysis. Additionally, it is important to note that – differently from the general trend in primates – all the species included in our sample express bi-parental care (i.e. care provided by the father and

the mother), meaning that there is a certain degree of allomaternal care even in non-cooperatively breeding, territorial corvid species. Future studies that more elaborately evaluate the degree of allo-maternal care in wild corvid populations are thus needed to create a comprehensive comparison between the corvids in this study and the primates (*Burkart et al., 2014*).

The evenness of access to food was medium to high in all tested species in our study (*Burkart et al., 2014*). Following the argument of the original study in primates (*Burkart et al., 2014*), that would indicate medium-to-high levels of social tolerance in all groups, irrespective of the prevalence of cooperative breeding or colonial nesting. However, the relatively even access to food among the group members may reflect that most corvids have a tendency to cache food for later consumption (*de Kort and Clayton, 2006*). In our study, even the most dominant birds rarely monopolized the apparatus for long durations, as it has been documented in despotic primate species (*Schnoell and Kappeler, 2018*). The corvids rather periodically left the area to cache their obtained food out of sight of their conspecifics (*Bugnyar et al., 2016*). Therefore, the evenness of access to food in phase II of the GSP might not be a valid proxy for social tolerance in corvids. Other approaches like co-feeding experiments might provide more suitable measures of social tolerance, because they measure how tolerant the individuals of a given group or species are to foraging in close proximity with other group members (*Sima et al., 2016*). Nevertheless, the use of the GSP has many advantages when attempting to conduct a comprehensive experimental investigation of pro-social behavior (*Burkart et al., 2014*): the apparatus and procedure are cognitively not demanding and testing individuals within their social groups and home environment reduces stress and increases animal welfare. Additionally, the paradigm offers several criteria to assess whether an individually was sufficiently habituated/trained and its propensity to land on the apparatus was not caused by a lack of inhibitory control. Overall, the birds tested in this study differentiated between the prosocial test and both control conditions and landed more often on the provisioning perch when they could provide food to their group members than when there was no food or when access to the food was blocked for the recipient (see *Appendix 1—figure 1*). Therefore, the GSP is a highly useful paradigm for comparative investigations of animal prosociality and can be conceivable applied to a much wider range of species and taxa.

The current study is a first attempt to determine how generalizable the predictions of the cooperative breeding hypothesis and self-domestication hypothesis are, or whether they are actually restricted to the primate order (*Thornton and McAuliffe, 2015*). In a systematic comparison of prosocial preferences across eight corvid species we find, in fact, evidence for both hypotheses. It is important to note, however, that these two hypotheses are not mutually exclusive and that one common underlying mechanism in both hypotheses is likely a heightened level of social tolerance at the nest. In cooperatively breeding species, helpers have to show increased social tolerance toward offspring that is not their own, while the breeders have to tolerate older offspring and immigrant helpers in their territories and close to their nests. In colonially nesting species, a breeding pair has to tolerate the proximity of other breeding pairs close to their nest. Consequently, the combined results of our study strongly suggest that both cooperative breeding and the heightened social tolerance required by colonial nesting are likely evolutionary pathways to prosocial behavior in corvids.

## Materials and methods

### Subjects

We tested 11 social groups of 8 corvid species (total N = 72 individuals: azure-winged magpies: group 1 N = 5, group 2 N = 4; carrion crows: N = 6; rooks: N = 12; Eurasian jackdaws: N = 14; New-Caledonian crows: group 1 N = 3, group 2 N = 2; common ravens: N = 9; large-billed crows: N = 9; Siberian jays: group 1 N = 5, group 2 N = 3; see *Appendix 2—table 1* for information on study sites, subject and husbandry details, and testing period for all study groups). We recruited and tested as many species and birds per species as possible, which resulted in the sample we describe here. Consequently, we did not perform any a priori sample size calculations. Biological replications could be performed for the three species for which we could test two independent social groups (i.e. azure-winged magpies, New-Caledonian crows, Siberian jays).

Besides Siberian jays, all species were tested in captivity, in their home aviary and social group prior to their first feeding of the day. High-quality food reward was used to encourage participation

in the experiment. The two Siberian jay groups were tested in the wild near the center of their territory. Here, less preferred food was provided near the apparatus to keep the group near the apparatus. The birds from all species were well habituated to participating in behavioral experiments (see Appendix 2 for habituation procedures and criteria).

### Ethical note

The study followed the Guidelines for the Use of Animals (*Vitale et al., 2018*), in accordance with national legislations. All animal care and data collection protocols were reviewed and approved by the ethical boards of the respective research institutions (see *Appendix 2—table 1*).

### Apparatus and procedure

We used the same apparatus with a seesaw mechanism as a previous study (*Horn et al., 2016*; *Figure 1a*), adjusted in size and weight to the different species. The apparatus consisted of a board outside the aviary, on which the food item was placed, and two sticks reaching through the wire mesh into the aviary on one side of the board with a provisioning perch fixed at their end. For the Siberian jays, the board was placed inside a wire mesh container, preventing individuals to access the board, but allowing them to freely access the provisioning perch on the outside. The apparatus' mechanism was balanced so that in the starting position the perch pointed up and the board pointed down. When a bird landed on the provisioning perch, its weight moved the seesaw down (*Figure 1a*). As soon as the bird left the perch, the apparatus automatically moved back to its original position. Near the other side of the board, inside the aviary, were perches that were not connected to the apparatus' seesaw mechanism. Food could be put on the board in two positions: one in front of the provisioning perch (Position 0) and one on the other side of the board (Position 1) out of reach from the perch. If food was placed in position 0, a subject could deliver food to itself by landing on the provisioning perch, after which the food slid toward the wire mesh and in reach. If food was placed in position 1 and a bird landed on the provisioning perch, it could not obtain the food itself. If it stayed on the perch long enough for another group member to arrive in position 1, it made food available to this group member (*Figure 1b*, *Video 1*). However, if the bird left the provisioning perch before another group member arrived, the apparatus moved back in the starting position and the food became unavailable. Therefore, multiple landings on the provisioning perch were possible within one trial.

We replicated the procedures of a previous study (*Horn et al., 2016*). The experiment consisted of six consecutive phases in a fixed sequence (three habituation/training phases and three test phases) and an additional retest phase for seven of the groups (*Figure 1c*; see Appendix 2 for detailed procedures).

In the access to food assessment (phase II) the apparatus' seesaw mechanism was fixed so that any bird landing in position one could obtain food. In two sessions, we placed food pieces sequentially in position one and recorded how many food pieces each group member obtained. In the group service test (phase IV), the seesaw mechanism was fully released and food was placed in position 1, so that a bird landing on the provisioning perch could only make food available to the group, not to itself (*Video 1*). On alternating days, we conducted empty control sessions, which were identical to test sessions except that no food was placed on the apparatus and therefore no food was available to be provided for the group members (*Video 2*). In the blocked control (phase V), access to food in position one was blocked with a fine net. Therefore, although food was visible, no food could be provided for the group members (*Video 3*). This was done to test whether landing was simply elicited by the presence of food. To ensure that the birds had comparable motivation levels (e.g. hunger) in all conditions, we conducted all sessions at the same testing times per day for each respective species. For the analysis of phases IV and V, we used only the summed data from the last two sessions (sessions 4 and 5) of each condition, because by then each bird had had the opportunity to learn about the consequences of operating the apparatus. The group service retest (phase VI) represents a technical replication and was identical to phase IV and consisted of two prosocial test and two empty control sessions. In all sessions of phases IV to VI, we interspersed motivation trials after every five regular trials where food was placed in position 0 to ensure that the birds were still motivated to participate in the experiment (see *Appendix 1—table 6*). We recorded how often each

individual landed on the provisioning perch during the regular trials. Additionally, we recorded which animal obtained the food and which animal provided the food in phases IV and VI.

## Data analysis

Providing/receiving of food and landings on the apparatus were scored live by the experimenter and confirmed via later video scoring. A second rater, who was not the experimenter for the respective group, scored the behavioral variables for 24% of all 270 test sessions, which included 50% of all sessions on which the main analyses were based. Inter-rater reliabilities were excellent across all groups (mean $ICC_{Group}$ ± SD = 0.975±0.041, minimum $ICC_{Group}$ = 0.878, maximum $ICC_{Group}$ = 0.998). All analyses were based on the data from the first rater. Results from the two groups of azure-winged magpies were previously reported in *Horn et al., 2016*.

For each group, we calculated the evenness of access to food in both sessions of phase II for those individuals that passed the habituation criterion in the preceding phase (total N = 63; see *Appendix 1—table 1* for details on each group). To calculate the evenness of access to food for each group (N = 11), we used Pielou's J' (*Pielou, 1977*) (i.e. an index ranging from 0, indicating maximal inequality to 1, indicating a completely equal distribution; see *Horn et al., 2016*) and calculated an averaged Pielou's J' across both test sessions (see *Source code 1*, part one for details). Further, we calculated the percentage of provided food in the last two sessions of phase IV. The percentage of provided food was corrected by including only provisioning by individuals that passed the criterion of landing significantly more often in the test compared to both control conditions (see *Burkart et al., 2014*), thereby giving the most conservative measure of prosocial provisioning. The rate of individuals that passed this criterion ranged from 0% to 50% across species (see *Appendix 1—table 1* for details). Note, however, that raw measures (including all birds) and corrected measures (including only birds that met the criterion) of food provisioning were highly correlated (Spearman's rho = 0.892, p≤0.001, N = 11 groups).

Since successful food provisioning in the GSP depended not only on a subject's landing on the apparatus, but also on the temporal and spatial coordination between donor and recipient, we could not exclude that a lack of coordination prevented food provisioning in some cases. Therefore, to further investigate the influence of cooperative breeding and colonial nesting on prosocial tendencies, we used the sum of the number of landings in the last two sessions of the prosocial test (phase IV) of all birds that passed the training criterion in the preceding phase (see *Appendix 1—table 1*). We had to additionally exclude four birds with unknown sex from this analysis, resulting in a total sample size of 51 birds. Only one data point per individual was used in all statistical analyses.

In a first step, we calculated a linear mixed-effects model (maximum likelihood method; package *lme4*; *Bates et al., 2015*) with 'number of landings in the prosocial test' as response variable, 'cooperative breeding', 'nesting type', 'sex', and all possible interactions as factors, 'group size' as additional factor without interactions, and 'group ID' nested within 'species' as random factors (see *Source code 1*, part 2). The variance of the random factors 'group ID' and 'species' was zero, resulting in a singular fit of the model. Therefore, we decided to calculate a general linear model with the same response variable and factors, but excluding 'group ID' and 'species' as random factors (see *Source code 1*, part 3). Note, however, that the results of both models are equivalent (see *Appendix 1—table 2*). We then obtained the candidate set of models by using the function *dredge* of the package *MuMIn* (*Bartoń, 2009*) to derive all possible sub-models with all possible combinations from the set of factors (including the intercept-only model) ranked by AICc (*Hurvich and Tsai, 1989*). Next, we selected the top 2AICc models (i.e. all models with a delta AICc ≤2 compared to the best-fitting model [*Burnham et al., 2002*]; 2 out of 256 models) and averaged them using the function *model.avg* in the package *MuMIn* (see *Source code 1*, part 3 for complete R script of this procedure). The intercept-only model did not fall within the range of top 2AICc models (delta AICc = 5.80). The factor 'group size' was not present in the final selection of best-fitting models (*Figure 2—source data 1*). *Figure 2—source data 2* shows the estimates and their standard errors (SE), z-values, sum of AICc weights, and number of models containing the specific factor of the averaged model. Factors with a sum of AICc weights larger than 0.5 and whose SE of the estimates did not overlap 0 were considered to have a high explanatory degree. The quality of all models was confirmed by investigating Q-Q plots and testing the normal distribution of the residuals. To ensure that choosing a threshold of delta AICc ≤2 did not lead to the exclusion of any potentially important factors (e.g. group size) we re-did the model selection and averaging procedure with a

threshold of delta AICc ≤7 (see *Source code 1*, part 3). This model included three additional factors, but all three had only minimal explanatory degree (i.e. the interaction between cooperative breeding and nesting type, the factor group size, the three-way interaction), thereby supporting original threshold of delta AICc ≤2 (see *Appendix 1—table 3* for detailed results with top AICc7 models). We used the same procedure as described above when analyzing the data separately for the females and for the males (see *Source code 1*, part 4). We decided not to include 'group size' into these models because of the small sample size and because 'group size' did not emerge as an important predictor for the complete dataset. Again, the variance of the random factors 'group ID' and 'species' was zero and we decided to calculate linear models, with the same response variable and predictors, excluding 'group ID' and 'species' as random factors (see *Source code 1*, part 4). We then derived all possible submodels from this set of predictors. For the female birds, there were two top 2AICc models and for the male birds, there were also two top 2AICc models (out of 8 models each). Full results of the two averaged models can be seen in *Figure 2—source data 2*. For testing the robustness of our model with the complete dataset, as well as the single sex models, we used the same procedure as described for the full dataset, while always excluding one species at a time (see *Source code 1*, part 5; see Appendix 1 for results on the single sex models; see *Appendix 1—table 4* for the detailed results excluding the Siberian jays). Additionally, we used one-sided Welch t-tests to test whether we could find the predicted significant difference when only testing for the effect of nesting type in the males and for the effect of cooperative breeding in the females, respectively. For testing whether there was a difference in the number of landings on the provisioning perch among only these individuals that passed the criterion of landing significantly more often in the test compared to both control conditions, due to the small sample size (N = 12) we used non-parametric Mann-Whitney U tests separately for the factors 'cooperative breeding' and 'nesting type'.

To test the extent to which common ancestry affected the birds' prosocial tendencies, we used the packages *geiger* (*Harmon et al., 2008*) and *MCMCglmm* (*Hadfield, 2010*) to calculate a phylogenetically controlled mixed-effects model with 'number of landings in the prosocial test' as response variable, and those parameters that were present in the top 2AICc models of the original analysis (i.e. 'cooperative breeding', 'nesting type', 'sex', and the interactions between 'cooperative breeding' and 'sex' and 'nesting type' and 'sex'). Additionally, we added 'phylogenetic effect' and 'species' as random effects. We further calculated the posterior mean (mean of the posterior distribution), the posterior mode (most likely value regarding the posterior distribution) and the 95% credible interval of the phylogenetic signal λ (see *Source code 1*, part 6 for complete R script).

Finally, to investigate whether opposite-sex provisioning occurred more often than same-sex provisioning, we calculated a non-parametric Wilcoxon signed-rank test (two-tailed). For each of the phases, we included only those individuals that had reached the respective habituation/training criterion (see Appendix 2 for details on the criteria). All statistical tests were carried out in R version 3.6.0 (2019-04-26). *Figure 2* and *Appendix 1—figure 1* were created with the package *ggplot2* (*Wickham, 2016*).

## Acknowledgements

This study was supported by the Austrian Science Fund (FWF; P26806 to JJMM; Y366-B17 to TB), the Vienna Science and Technology Fund (WWTF; CS11-008 to TB), the ERA-Net BiodivERsA (31BD30_172465 to MG), the University of Vienna (Marie Jahoda grant to LH; Förderungsstipendium to MH and CR; Uni:Docs doctoral fellowship to L-CV), the JSPS KAKENHI (17H02653, 16H06324 to E-II; 15J02148 to MSu), the JST CREST (JPMJCR17A4 to E-II), the Keio University ICR Projects (MKJ1905 to E-II), a Royal Society of New Zealand Rutherford Discovery Fellowship (AHT), and a Prime Minister's McDiarmid Emerging Scientist Prize (AHT).

We thank Nadja Kavcik-Graumann for drawing the illustrations in *Figure 1*, Sarah Vlasitz for her help with habituating the ravens, and Hans Frey for granting access to the rooks at the Eulen- und Greifvogelstation Haringsee. Further, we thank Province Sud for granting us permission to work in New Caledonia, Dean M and Boris C for allowing us access to their properties for catching and releasing the crows, Russel Gray for granting access to the New Caledonian Crow Lab at the University of Auckland, and Romana Gruber for her help with reliability coding. Finally, we are grateful to András Péter for constructing the apparatuses and the animal care staff at all involved research facilities.

## Additional information

### Funding

| Funder | Grant reference number | Author |
|---|---|---|
| Austrian Science Fund | P26806 | Jorg JM Massen |
| Austrian Science Fund | Y366-B17 | Thomas Bugnyar |
| Vienna Science and Technology Fund | CS11-008 | Thomas Bugnyar |
| ERA-Net BiodivERsA | 31BD30_172465 | Michael Griesser |
| University of Vienna | Förderungsstipendium | Marietta Hengl<br>Christiane Rössler |
| University of Vienna | Uni:Docs doctoral fellowship | Lisa-Claire Vanhooland |
| JSPS | KAKENHI 17H02653 | Ei-Ichi Izawa |
| JSPS | KAKENHI 16H06324 | Ei-Ichi Izawa |
| JSPS | KAKENHI 15J02148 | Masaki Suyama |
| JST | CREST JPMJCR17A4 | Ei-Ichi Izawa |
| Keio University | ICR Projects MKJ1905 | Ei-Ichi Izawa |
| Royal Society of New Zealand | Rutherford Discovery Fellowship | Alex H Taylor |
| Prime Minister's McDiarmid Emerging Scientist Prize | | Alex H Taylor |
| University of Vienna | Marie Jahoda grant | Lisa Horn |

The funders had no role in study design, data collection and interpretation, or the decision to submit the work for publication.

### Author contributions

Lisa Horn, Conceptualization, Data curation, Formal analysis, Supervision, Validation, Investigation, Visualization, Methodology, Writing - original draft, Project administration, Writing - review and editing; Thomas Bugnyar, Jorg JM Massen, Conceptualization, Supervision, Funding acquisition, Methodology, Writing - review and editing; Michael Griesser, Formal analysis, Funding acquisition, Writing - review and editing; Marietta Hengl, Masaki Suyama, Data curation, Funding acquisition, Investigation; Ei-Ichi Izawa, Christiane Rössler, Data curation, Funding acquisition, Investigation, Writing - review and editing; Tim Oortwijn, Data curation, Investigation, Writing - review and editing; Clara Scheer, Conceptualization, Data curation, Investigation, Methodology; Martina Schiestl, Yvonne Zürcher, Data curation, Investigation; Alex H Taylor, Auguste MP von Bayern, Funding acquisition, Writing - review and editing; Lisa-Claire Vanhooland, Funding acquisition, Validation, Investigation

### Author ORCIDs

Lisa Horn (iD) https://orcid.org/0000-0002-9586-915X
Michael Griesser (iD) http://orcid.org/0000-0002-2220-2637

### Ethics

Animal experimentation: The study followed the Guidelines for the Use of Animals (Vitale et al., 2018), in accordance with national legislations. All animal care and data collection protocols were reviewed and approved by the ethical boards of the respective research institutions (see Appendix 2-table 1).

### Decision letter and Author response

Decision letter https://doi.org/10.7554/eLife.58139.sa1

Author response https://doi.org/10.7554/eLife.58139.sa2

## Additional files

### Supplementary files

• Source code 1. Source code for the calculation of Pielou's J' and for running the statistical models in R.

• Transparent reporting form

### Data availability

The datasets analyzed in this study are available on Dryad.

The following dataset was generated:

| Author(s) | Year | Dataset title | Dataset URL | Database and Identifier |
|---|---|---|---|---|
| Horn L, Bugnyar T, Griesser M, Hengl M, Izawa E-I, Oortwijn T, Rössler C, Scheer C, Schiestl M, Suyama M, Taylor AH, Vanhooland L-C, Bayern AMP, Zürcher Y, Massen JJM | 2020 | Sex-specific effects of cooperative breeding and colonial nesting on prosociality in corvids | http://dx.doi.org/10.5061/dryad.s7h44j14d | Dryad Digital Repository, 10.5061/dryad.s7h44j14d |

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

# Appendix 1

## Supplementary results
### Number of individuals reaching the habituation/training criteria

For each group, *Appendix 1—table 1* shows the number of individuals that reached the criterion for being included in the analysis of the access to food assessment (phase II), the criterion for being included in the analysis of the prosocial test (phase IV), and the criterion for being included in the provisioning data.

**Appendix 1—table 1.** Group size and number of individuals passing the selection criteria across all tested groups.

Given are – for each group – the group size and the number of birds passing the criteria for phases II and IV (i.e. taking at least 10 pieces of food in a minimum of five sessions in the previous phase) and the criterion of landing in significantly more trials in the prosocial test than in both the empty and the blocked control (Fisher's exact test).

| Species | Group | Group size | Criterion Phase II | | Criterion Phase IV | | Criterion test vs. controls |
|---|---|---|---|---|---|---|---|
| | | | N | Sessions, median (min, max) | N | Sessions, median (min, max) | |
| Azure-winged magpies | 1 | 5 | 5 | 9 (6, 11) | 5 | 23 (22, 30) | 3 |
| | 2 | 4 | 4 | 7.5 (5, 12) | 3 | 33 (33, 49) | 1 |
| Carrion crows | 1 | 6 | 6 | 9 (5, 13) | 6 | 11 (9, 17) | 2 |
| Rooks | 1 | 12 | 5 | 10 (5, 17) | 3 | 65 (65, 65) | 1 |
| Eurasian jackdaws | 1 | 14 | 12 | 26 (5, 51) | 10 | 59 (58, 65) | 2 |
| New-Caledonian crows | 1 | 3 | 3 | 8 (8, 8) | 3 | 17 (16, 18) | 1 |
| | 2 | 2 | 2 | 5 (5, 5) | 2 | 13 (12, 14) | 0 |
| Common ravens | 1 | 9 | 9 | 7 (5, 26) | 7 | 30 (27, 37) | 1 |
| Large-billed crows | 1 | 9 | 9 | 5 (5, 6) | 9 | 10 (10,12) | 1 |
| Siberian jays | 1 | 5 | 5 | 5 (5, 9) | 4 | 5 (5, 5) | 0 |
| | 2 | 3 | 3 | 5 (5, 5) | 3 | 10 (10, 10) | 0 |

### Full model including 'group ID' and 'species' as random factors

The full linear mixed-effects model (maximum likelihood method) used 'number of landings in the prosocial test' as response variable, 'cooperative breeding', 'nesting type', 'sex', and all possible interactions as predictors, 'group size' as additional predictor without interactions, and 'group ID' nested within 'species' as random factors (see *Source code 1*, part 1). Variance of the random factors 'group ID' and 'species' was found to be zero, resulting in a singular fit of the model. We then derived all possible submodels from the set of predictors (including the intercept-only model), selected the top 2AICc of models (2 out of 256 models), and averaged them using the *model.avg* function in the *MuMIn* package in R (see *Source code 1*, part 1). The predictor 'group size' was not present in the final selection of best-fitting models. *Appendix 1—table 2* shows the estimates, conditional standard errors (SE), confidence intervals, z-values, p-values, and relative importance of the averaged model.

**Appendix 1—table 2.** Effects of cooperative breeding, nesting type and sex on the number of landings in the prosocial test.

Given are estimates, standard errors (SE), z-values, sum of AICc weights (SW$_{AICc}$), and number of models containing the specific factor (N$_{Models}$) after model averaging. Factors with a sum of AICc weights larger than 0.5 and whose SE of the estimates did not overlap 0 were considered to have a high explanatory degree and are given in bold. Number of individuals: N = 51.

| Parameter | Estimate | SE | Z | SW_AICc | N_Models |
|---|---|---|---|---|---|
| (Intercept) | 6.403 | 3.739 | 1.675 | - | - |
| Cooperation (yes) | 10.024 | 4.226 | 2.304 | 0.61 | 1 |
| Nesting (territorial) | 4.092 | 4.073 | 0.977 | 1.00 | 2 |
| Sex (male) | 17.075 | 5.672 | 2.947 | 1.00 | 2 |
| Cooperation (yes) x Sex (male) | −16.057 | 6.445 | 2.420 | 0.61 | 1 |
| Nesting (territorial) x Sex (male) | −19.611 | 6.014 | 3.172 | 1.00 | 2 |

## Full model with the top 7AICc of models

The full linear model used 'number of landings in the prosocial test' as response variable, 'cooperative breeding', 'nesting type', 'sex', and all possible interactions as predictors, and 'group size' as additional predictor without interactions (see *Source code 1*, part 1). We then derived all possible submodels from the set of predictors (including the intercept-only model), selected the top 7AICc of models (14 out of 256 models), and averaged them using the *model.avg* function in the *MuMIn* package in R (see *Source code 1*, part 1). The intercept-only model fell within the range of top 7AICc models (delta AICc = 5.80). *Appendix 1—table 3* shows the estimates, conditional standard errors (SE), confidence intervals, z-values, p-values, and relative importance of the averaged model.

**Appendix 1—table 3.** Threshold for model selection and averaging set to delta AICc ≤ 7.
Given are estimates, standard errors (SE), z-values, sum of AICc weights (SW_AICc), and number of models containing the specific factor (N_Models) after model averaging. Factors with a sum of AICc weights larger than 0.5 and whose SE of the estimates did not overlap 0 were considered to have a high explanatory degree and are given in bold. Number of individuals: N = 51.

| Parameter | Estimate | SE | Z | SW_AICc | N_Models |
|---|---|---|---|---|---|
| (Intercept) | 6.836 | 5.400 | 1.240 | - | - |
| **Cooperation (yes)** | 9.201 | 5.877 | 1.534 | 0.73 | 10 |
| Nesting (territorial) | 3.974 | 4.859 | 0.799 | 0.97 | 12 |
| **Sex (male)** | 17.192 | 5.954 | 2.829 | 0.93 | 10 |
| **Cooperation (yes) x Sex (male)** | −16.199 | 7.035 | 2.240 | 0.59 | 5 |
| **Nesting (territorial) x Sex (male)** | −19.698 | 6.429 | 2.985 | 0.93 | 10 |
| Cooperation (yes) x Nesting (territorial) | −3.984 | 7.676 | 0.505 | 0.17 | 4 |
| Group size | −0.154 | 0.594 | 0.253 | 0.19 | 4 |
| Cooperation (yes) x Nesting (territorial) x Sex (male) | 0.902 | 14.106 | 0.062 | 0.02 | 1 |

## Evaluating the robustness of the models

For testing the robustness of our model with the complete dataset, as well as the single sex models, we used the same procedure as described for the full dataset (see main document, section 'Data analysis') with reduced datasets in which we always excluded one species at a time. The full linear model used 'number of landings in the prosocial test' as response variable, 'cooperative breeding', 'nesting type', 'sex', and all possible interactions as predictors, and 'group size' as additional predictor without interactions (see *Source code 1*, part 5). For each reduced dataset, we then proceeded with model selection and averaging in the same way as we did in the original model.

Four out of eight models had the same results as before (i.e. the main factors sex and cooperative breeding, as well as the interactions between both cooperative breeding and nesting type with sex had a high explanatory degree; removed species: Siberian jays, N = 48; rooks, N = 48; common ravens, N = 44; carrion crows, N = 45), while the main factor nesting type had an added high explanatory degree in two models (removed species: New-Caledonian crows, N = 46; azure-winged magpies, N = 43). In one model nesting type, sex, and the interaction between these two factors

had a high explanatory degree, while cooperative breeding and the interaction between cooperative breeding and sex were only marginally important (i.e. $SW_{AICc}$ = 0.44; removed species: large-billed crows, N = 42). Finally, in one model the intercept-only model was included in the selection of best-fitting models (removed species: Eurasian jackdaws, N = 41), implying that the averaged model was not robust.

Siberian jays were the only species tested in the wild. In the prosocial test, they did not manage to coordinate to successfully provide food to their group members. However, even when excluding the Siberian jays from the dataset, the results were equivalent to the original model (*Appendix 1—table 4*). This confirms that our results were not driven by the low number of landings in the wild population per se.

**Appendix 1—table 4.** Effects of cooperative breeding, nesting type and sex on the number of landings in the prosocial test without the Siberian jays.
Given are estimates, standard errors (SE), z-values, sum of AICc weights ($SW_{AICc}$), and number of models containing the specific factor ($N_{Models}$) after model averaging. Factors with a sum of AICc weights larger than 0.5 and whose SE of the estimates did not overlap 0 were considered to have a high explanatory degree and are given in bold. Number of individuals: N = 48.

| Parameter | Estimate | SE | Z | $SW_{AICc}$ | $N_{Models}$ |
|---|---|---|---|---|---|
| (Intercept) | 6.105 | 3.989 | 1.495 | - | |
| **Cooperation (yes)** | **10.317** | **4.637** | **2.161** | **0.67** | **1** |
| Nesting (territorial) | 3.835 | 4.439 | 0.839 | 1.00 | 2 |
| **Sex (male)** | **17.640** | **6.059** | **2.848** | **1.00** | **2** |
| **Cooperation (yes) x Sex (male)** | **−16.783** | **7.064** | **2.308** | **0.67** | **1** |
| **Nesting (territorial) x Sex (male)** | **−18.678** | **6.574** | **2.763** | **1.00** | **2** |

When using reduced datasets that included only the male birds, all eight models had the same results as before (i.e. only the main factor nesting type had a high explanatory degree; removed species: Eurasian jackdaws, N = 20; Siberian jays, N = 23; rooks, N = 24; common ravens, N = 23; New-Caledonian crows, N = 21; large-billed crows, N = 20; carrion crows, N = 23; azure-winged magpies, N = 21). Additionally, the male birds from colonial species landed significantly more often on the provisioning perch than the male birds from territorial species, when only testing for the factor nesting type (Welch t-test: t = 3.01, df = 13.66, p-value=0.005).

When using reduced datasets that included only the female birds, only two out of eight models had the same results as before (i.e. only the main factor cooperative breeding had a high explanatory degree; removed species: Siberian jays, N = 25; common ravens, N = 21), while nesting type had an added high explanatory degree in one model (removed species: azure-winged magpies, N = 22). In five models, the intercept-only model was included in the selection of best-fitting models (removed species: Eurasian jackdaws, N = 21; rooks, N = 24; New-Caledonian crows, N = 25; large-billed crows, N = 22; carrion crows, N = 22), implying that the averaged models were not robust. Also when testing only whether the females from cooperatively breeding species landed more often on the provisioning perch than the females from species that do not breed cooperatively, the results were only marginally significant (Welch t-test: t = −1.64, df = 8.30, p-value=0.069).

## Phylogenetically controlled model

To test the extent to which common ancestry affected the birds' prosocial tendencies, we calculated a phylogenetically controlled mixed-effects model with 'number of landings in the prosocial test' as response variable, and those parameters that were present in the top 2AICc models of the original analysis (i.e. 'cooperative breeding', 'nesting type', 'sex', and the interactions between 'cooperative breeding' and 'sex' and 'nesting type' and 'sex'). Additionally, we added 'phylogenetic effect' and 'species' as random effects (see *Source code 1*, part 6). The results were equivalent to the original model: the main factors cooperative breeding and sex significantly predicted the number of landings on the provisioning perch in the prosocial test, and these main effects were again qualified by

significant interactions between both cooperative breeding and sex and nesting type and sex (*Appendix 1—table 5*).

**Appendix 1—table 5.** Effects of cooperative breeding, nesting type, and sex on the number of landings in the prosocial test in a phylogenetically controlled model.
Given are the posterior mean of the estimate (Post. mean), its 95% credible interval (95% HPD interval), its effective sample size (Eff. samp.), and p-value ($P_{MCMC}$) of each parameter. Number of individuals: N = 51. *p≤0.05, **p≤0.01, ***p≤0.001.

| Parameter | Post. mean | 95% HPD interval | Eff. samp. | $P_{MCMC}$ |
|---|---|---|---|---|
| (Intercept) | 5.012 | [−3.198, 13.247] | 10111 | 0.210 |
| Cooperation (yes) | 10.001 | [0.082, 19.886] | 9998 | 0.048* |
| Nesting (Territorial) | 4.346 | [−5.408, 13.376] | 9998 | 0.347 |
| Sex (male) | 19.660 | [8.899, 30.292] | 9998 | 0.0002*** |
| Cooperation (yes) x Sex (male) | −20.576 | [−33.588,−8.551] | 9998 | 0.002** |
| Nesting (territorial) x Sex (male) | −16.394 | [−30.183,−2.329] | 9998 | 0.020* |

## Comparison of food provisioning in the original prosocial test and re-test of the prosocial test

Food provisioning in the original prosocial test and the re-test of the prosocial test, which was conducted after the main experiment, was correlated both on the group level (Spearman, N = 7, rho = 0.821, p=0.023) and on the individual level (N = 43, rho = 0.385, p=0.011).

## Comparison of the number of landings on the provisioning perch in the different test phases

Across all species and groups (N = 55 birds), the birds differentiated between the prosocial test, the empty control, and the blocked control (Kruskal-Wallis chi-squared = 11.6, df = 2, p=0.003). They landed on the provisioning perch more often in the prosocial test (mean = 11.3, median = 6, min = 0, max = 46) than in both the empty control (mean = 5.0, median = 3, min = 0, max = 24; Wilcoxon W = 1010.5, p=0.003) and the blocked control (mean = 5.5, median = 2, min = 0, max = 45; Wilcoxon W = 994.5, p=0.002). There was no difference between the empty and the blocked control (Wilcoxon W = 1453.5, p=0.723).

When considering the groups with which we conducted the re-test (N = 43 birds), we found that these birds landed on the provisioning perch more often in the repeated prosocial test (mean = 10.1, median = 6, min = 0, max = 47) than in the repeated empty control (mean = 2.4, median = 1, min = 0, max = 10; Wilcoxon W = 547.5, p=0.001). There was no difference between the number of landings in the original prosocial test (mean = 12.7, median = 7, min = 0, max = 46) and in the re-test of the prosocial test (Wilcoxon W = 781.5, p=0.216).

*Appendix 1—figure 1* shows the number of landings in the prosocial test, the empty control, and the blocked control, split by species. Additionally, for the six species with which we repeated the prosocial test and the empty control, *Appendix 1—figure 1* shows the number of landings in the re-test.

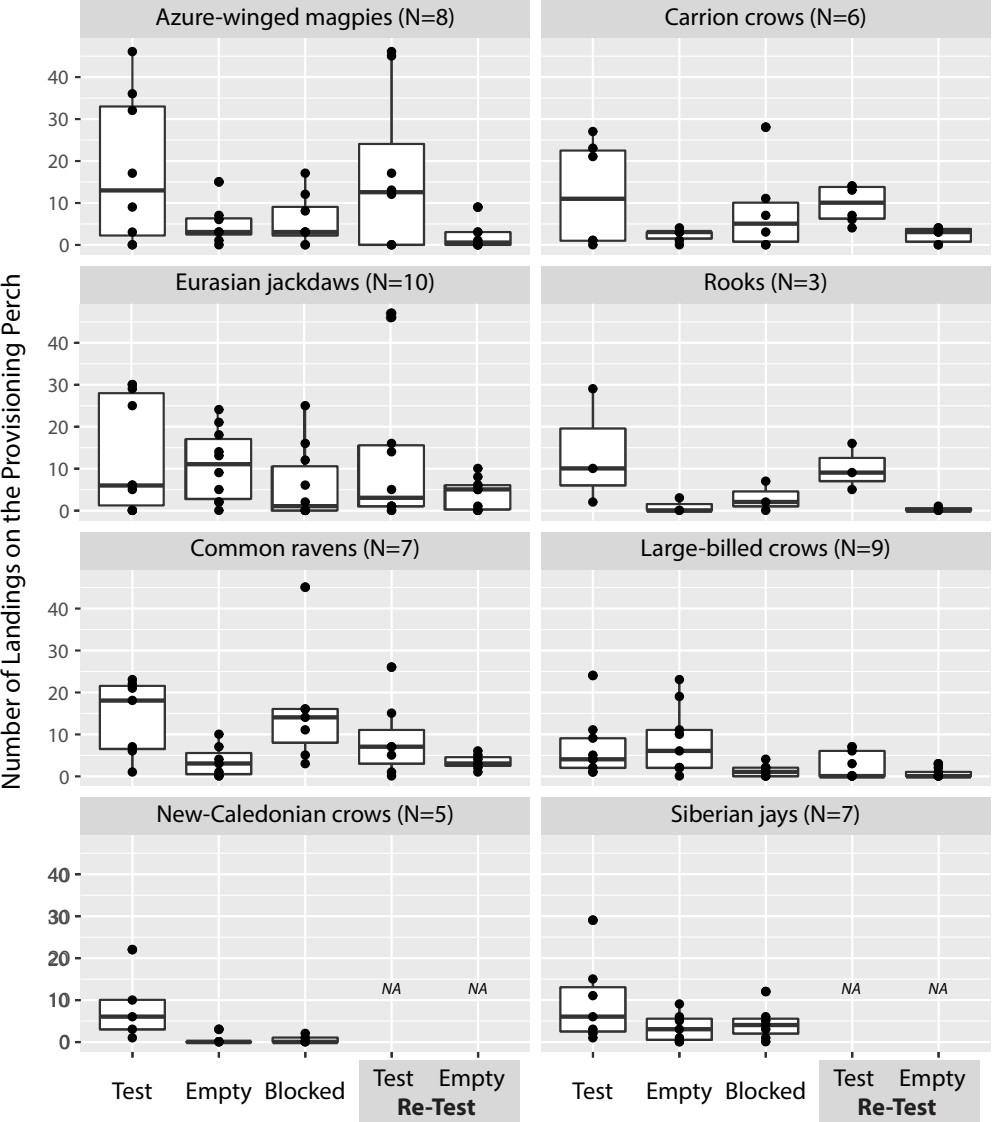

**Appendix 1—figure 1.** Number of landings in all test phases, split by species. The box plots represent medians (horizontal lines), inter-quartile ranges (boxes), as well as minima, maxima (whiskers). All data are represented with dots. Dots not encompassed by the whiskers are outliers. NA = not available.

## Comparison of the percentage of landings in the motivation trials of the different test phases

Across all groups (N = 11) there was no difference between the percentages of motivation trials in which any bird landed on the provisioning perch across conditions (Kruskal-Wallis chi-squared = 1.1, df = 4, p=0.892). *Appendix 1—table 6* shows the percentage of motivation trials with landings in the prosocial test, the empty control, and the blocked control, split by species and group. Additionally, for the six species with which we repeated the prosocial test and the empty control, *Appendix 1—table 6* shows the percentage of motivation trials with landings in the re-test.

**Appendix 1—table 6.** Percentage of motivation trials with landings.
Given are – for each group – the number of motivation trials in the last two sessions of each condition and the percentage of motivation trials in which any bird landed on the provisioning perch, as well as the median, minimum and maximum of these percentages. NA = not available, min = minimum, max = maximum.

| Species | Group | Motivation trials (N) | Test | Empty | Blocked | Re-test Test | Re-test Empty |
|---|---|---|---|---|---|---|---|
| Azure-winged magpies | 1 | 12 | 100 | 100 | 100 | 100 | 100 |
| | 2 | 12 | 92 | 100 | 100 | 100 | 100 |
| Carrion crows | 1 | 14 | 100 | 100 | 100 | 93 | 100 |
| Rooks | 1 | 14 | 100 | 100 | 79 | 93 | 93 |
| Eurasian jackdaws | 1 | 26 | 100 | 100 | 100 | 100 | 100 |
| Common ravens | 1 | 20 | 100 | 100 | 100 | 100 | 100 |
| Large-billed crows | 1 | 20 | 100 | 100 | 80 | 100 | 55 |
| New-Caledonian crows | 1 | 8 | 100 | 100 | 100 | NA | NA |
| | 2 | 6 | 83 | 83 | 100 | NA | NA |
| Siberian jays | 1 | 12 | 75 | 83 | 83 | NA | NA |
| | 2 | 8 | 75 | 88 | 50 | NA | NA |
| | | Median (min, max) | 100 (75, 100) | 100 (83, 100) | 100 (50, 100) | 100 (93, 100) | 100 (55, 100) |

## Appendix 2

### Supplementary methods

#### Supplementary procedure

The experiment consisted of six consecutive phases in a fixed sequence: three habituation/training phases and three test phases (see *Figure 1c*). With six species (seven groups) we repeated the test and empty control in an additional phase 1–3 months after the original test sessions to exclude that reduced landings in the blocked control were due to order effects (*Figure 1c*). None of the groups received any training with the apparatus in-between the original test and the re-test.

Due the groups' greatly different group sizes, we adjusted the number of trials per session in all phases to the number of individuals in the group. Like this, each individual in each group had an equal chance to obtain food rewards.

#### Phase 0 – Habituation to the apparatus

The apparatus was installed in the home aviary. After two weeks the seesaw mechanism was fixed with the provisioning perch pointing downwards. A food bowl was mounted in front of the perch on the inside of the aviary (Position 0; *Figure 1b*). In each session, the bowl was filled with highly preferred food (chosen depending on each species' preferences) and the birds were video-recorded for thirty minutes. A bird reached criterion when it had landed on the perch and fed from the bowl at least five times.

#### Phase I – Habituation to the procedure

The seesaw mechanism was still fixed with the perch in a downward position, so that a piece of food placed on the board would automatically slide to the wire mesh and into the birds' reach. On alternating days, pieces of food were provided either in position 0 or 1 (*Figure 1b*). In each trial, the experimenter called the birds' attention and placed one piece of food on the board. The next trial started after a bird obtained the food or after a maximum of 2 minutes. If a bird took the piece of food, the experimenter placed the next piece of food on the board. If no bird took the food, the experimenter called the birds' attention again, lifted the same piece of food and placed it back on the board. A session ended after number of individuals (N)*5 trials or when none of the birds landed on the perch for three consecutive trials. If a bird (or several birds) started monopolizing the apparatus, this bird (these birds) was (were) distracted or temporarily separated from the group. A bird reached the habituation criterion when it had taken at least 10 pieces of food in a minimum of 5 sessions. Average number of sessions to criterion, split by group can be seen in *Appendix 1—table 1*.

#### Phase II – Access to food assessment (test phase)

The seesaw mechanism was still fixed with the perch in a downward position. The experimenter put N*5 pieces of food on the apparatus in position 1, one at a time and called the birds' attention each time. After the food was taken the experimenter placed the next piece of food on the board. Two sessions of the access to food assessment were conducted on 2 consecutive days. We recorded how many pieces of food each bird obtained.

#### Phase III – Training

In this phase, the birds learnt to move food towards the wire mesh by landing on the perch. Food was always placed in position 0. To facilitate learning, the seesaw mechanism was first partially released – so that the perch moved only slightly – and food was placed close to the wire mesh. When each bird had obtained food from the apparatus at least once, the mechanism was released further. In the final step, the seesaw mechanism was completely released and the food was placed at the other end of the board.

In each trial, the experimenter called the birds' attention and placed one piece of food on the board. The next trial started after a bird obtained the food or after a maximum of 2 minutes. A session ended after N*5 trials or when none of the birds landed on the perch for three consecutive

trials. Again, if a bird started monopolizing the apparatus, this bird was distracted or temporarily separated from the group. A bird reached training criterion when it had taken at least 10 pieces of food in a minimum of 5 sessions with the seesaw mechanism completely released. Average number of sessions to criterion, split by group can be seen in *Appendix 1—table 1*.

### Phase IV – Group service (test phase)

In this phase, the apparatus' seesaw mechanism was completely released. We conducted five test sessions and five empty control sessions on alternating days. To ensure that the birds had comparable motivation levels (e.g. hunger) in all conditions we conducted all sessions at the same testing times per day for each respective species. Unforeseeable surrounding circumstances (e.g. bad weather) happened equally for the different conditions and did not seem to affect the birds' motivation to participate.

In a regular trial of a test session, a piece of food was placed in position 1 (see *Video 1*). Additionally, each session comprised motivation trials with food in position 0 in the very beginning of the session and after every fifth regular trial. Each session consisted of N*5 regular and N+1 motivation trials. In each trial the experimenter called the birds' attention and placed one piece of food on the board. The next trial started after a bird obtained the food or after a maximum of 2 minutes.

The empty control sessions were identical to the test sessions, except that in the regular control trials no food was placed on the board. In these trials, the experimenter approached the apparatus and pretended to leave a piece of food in position 1, while calling the birds' attention (see *Video 2*). Control sessions also comprised motivation trials with food in position 0. Each session consisted of N*5 regular and N+1 motivation trials.

For each trial, we recorded which animal(s) landed on the perch in position 0 (i.e. moved the seesaw mechanism) and which animal(s) landed in front of position 1. Additionally, we recorded which animal obtained the piece of food and which animal provided the piece of food.

### Phase V – Blocked control (test phase)

In this phase, the access to position one was blocked with a fine-meshed net, so that no food could be obtained in this position. Otherwise, the procedure was exactly the same as in group service and we conducted five blocked control sessions (i.e. food is placed in position 1; see *Video 3*) and five blocked empty control sessions (i.e. no food is placed in position 1) on alternating days. To ensure that the birds had comparable motivation levels (e.g. hunger) compared to the previous conditions we conducted all sessions at the same testing times per day as previously for each respective species. For each trial, we recorded which animal(s) landed on the perch in position 0 and which animal(s) landed in front of position 1.

### Phase VI – Group service re-test (test phase)

The procedure was exactly the same as in group service and we conducted two test sessions and two empty control sessions on alternating days. For each trial, we recorded which animal(s) landed on the perch in position 0 (i.e. moved the seesaw mechanism) and which animal(s) landed in front of position 1. Additionally, we recorded which animal obtained the piece of food and which animal provided the piece of food.

## Supplementary subject information

*Appendix 2—table 1* shows the study sites, subject and husbandry details, testing period, and ethical approval information for all study groups.

**Appendix 2—table 1.** Study sites, subject and husbandry details, testing period, and ethical approval information.

| Species | Study site | Subject and husbandry details | Testing period | Ethical approval |
|---|---|---|---|---|
| Azure-winged magpies (*Cyanopica cyana*) Group 1 | Haidlhof Research Station, University of Vienna and University of Veterinary Medicine Vienna, Austria | *Subjects:* two females, three males; all birds were adults and parent-raised. *Housing:* outdoor aviary (5 × 3×3 m), partially covered with a semi-transparent roof; the aviary used fine-grained sand as substrate and was equipped with fixed and swinging branches, live plants, stones, woodchips and gravel for caching food, a birdbath, and other enrichment objects. *Feeding:* the birds were fed daily with different fruits, insects, and seeds; water and pellets ('Beo komplet', NutriBird) were provided ad libitum; vitamin supplements and meat or egg were provided every second week. | Apr – Nov 2015; re-test: Apr 2016 | All animal care and data collection protocols were approved by the Animal Welfare Board of the Faculty of Life Sciences, University of Vienna (permit no. 2016–008). |
| Azure-winged magpies (*Cyanopica cyana*) Group 2 | Animal Care Facility of the Department of Cognitive Biology, University of Vienna, Austria | *Subjects:* two adult females, one adult male, one juvenile female (<1 year old); all birds were parent-raised. one additional juvenile bird was housed in the same aviary, but never participated in the experiment due to physical impairments. *Housing:* outdoor aviary (6 × 3×3 m), fully covered with a semi-transparent roof; for equipment see group 1. *Feeding:* see group 1 | Nov 2015 – Apr 2016; re-test: May 2016 | All animal care and data collection protocols were approved by the Animal Welfare Board of the Faculty of Life Sciences, University of Vienna (permit no. 2016–008). |

*Continued on next page*

*Appendix 2—table 1 continued*

| Species | Study site | Subject and husbandry details | Testing period | Ethical approval |
|---|---|---|---|---|
| Carrion crows (*Corvus corone*) | Haidlhof research station, University of Vienna and university of veterinary medicine vienna, Austria | *Subjects:* four females, two males; all birds were adults and hand-raised. By appearance, the crows were either carrion crows or hybrids of carrion and hooded crows, reflecting the hybridization belt in Europe. Both species have highly similar life histories and are often considered to belong to one species complex (*Vijay et al., 2016*). *Housing:* the aviary comprised a large outdoor part (12 × 9 × 5 m) and two adjacent roofed experimental compartments (3 × 4 × 5 m each); the aviary used coarse sand as substrate and was equipped with fixed and swinging branches, live plants, stones, woodchips and gravel for caching food, several birdbaths, and other enrichment objects. *Feeding:* the birds were fed a diverse diet containing meat, milk products, cereal, vegetables, and fruit twice a day; water was provided ad libitum. | Oct 2015 – May 2016; re-test: Jul 2016 | All animal care and data collection protocols were approved by the Animal welfare board of the faculty of life sciences, University of Vienna (permit no. 2016–017). |
| Common ravens (*Corvus corax*) | Haidlhof Research Station, University of Vienna and University of Veterinary Medicine Vienna, Austria | *Subjects:* three adult birds (>4 years old; 1F/2M), six subadult birds (2 years old; 4F/2M); all birds were hand-raised. *Housing:* large outdoor aviary (15 × 15×5 m) that could be divided into several compartments; equipment see carrion crows. *Feeding:* see carrion crows. | May – Oct 2016; re-test: Nov 2016 | All animal care and data collection protocols were approved by the Animal Welfare Board of the Faculty of Life Sciences, University of Vienna (permit no. 2016–017). |

*Continued on next page*

*Appendix 2—table 1 continued*

| Species | Study site | Subject and husbandry details | Testing period | Ethical approval |
|---|---|---|---|---|
| Large-billed crows (*Corvus macrorhynchos*) | Tsukuba Field Station, Keio University, Japan | *Subjects:* nine sub-adult birds (all were 3 years old; 4F and 5M); all birds were parent-raised and born in the wild. They were caught as free-floating yearlings in the wild and group-housed thereafter. *Housing:* outdoor aviary (10 × 10 × 3 m) that could be divided four experimental compartments (5 × 5×3 m); the aviary used coarse sand as substrate and was equipped with large branches, a water pool for bathing and other enrichment objects. *Feeding:* Dairy diet consisted of dog food, meat, eggs, dried fruits. Water was available ad libitum. | May – Jul 2016; re-test: Dec 2016 | Animal Care and Use Committee of Keio University (no. 16059) |
| New-Caledonian crows (*Corvus moneduloides*) Group 1 | La Foa, Province Sud, New Caledonia | *Subjects:* two adult birds (>3 years old; 1F and 1M) and one juvenile bird (1 st year; M); family group; all were wild caught, temporarily housed and released in the wild. *Housing:* Crows were housed in an outdoors aviary for temporary behavioral research purposes before being released back into the wild. *Feeding:* Daily diet consisted of meat, dog food, eggs, and fresh fruit, with water available ad libitum. | Jun – Jul 2017 | University of Auckland Animal Ethics Committee (reference no. 001823). |
| New-Caledonian crows (*Corvus moneduloides*) Group 2 | La Foa, Province Sud, New Caledonia | *Subjects:* one adult bird (>3 years old; M) and one juvenile bird (1 st year; M); father and son dyad; both were wild caught, temporarily housed and released in the wild. *Housing:* see group 1 *Feeding:* see group 1 | May – Jul 2016 | University of Auckland Animal Ethics Committee (reference no. 001823). |

*Continued on next page*

*Appendix 2—table 1 continued*

| Species | Study site | Subject and husbandry details | Testing period | Ethical approval |
|---|---|---|---|---|
| Rooks (*Corvus frugilegus*) | 'Eulen- und Greifvogelstation', Haringsee, Austria | *Subjects:* 10 adult birds (5F/5M), two subadult birds (1F/1M); all birds were parent-raised and born in the wild. *Housing:* outdoor aviary (3.3 × 7.4 × 3.1 m) with a roofed platform (3.3 × 1.1 m); the aviary used soil and bark chips as substrate and was equipped with large branches, a water pool for bathing and other enrichment objects. *Feeding:* the birds were fed on a daily basis with cereals, dried mealworms, minced meat mixed with calcium carbonate and small pieces of scrambled eggs; water was provided ad libitum; nuts and chicks were provided several times a week. | May 2016 – Mar 2017; re-test: Jun 2017 | All animal care and data collection protocols were approved by the Animal Welfare Board of the Faculty of Life Sciences, University of Vienna (permit no. 2016–017). |
| Siberian jays (*Perisoreus infaustus*) Group 1 | Wild population, studied near Arvidsjaur, Swedish Lapland (65°40 N, 19°0 E) | *Subjects:* male breeder, two non-breeders born in spring 2017, two juveniles born in spring 2018; one subject did not participate in phase 4; all individuals are members of a wild group of Siberian jays, part of a long-term study on individually color-ringed Siberian jays (see Ekman & Griesser 2016). *Living area:* The study was carried out in a natural setting in a wild population, thus the birds required no care. The apparatus was placed within the focal group's territory. We provided less preferred food (pig fat) on a standardized feeding device on the side of the experimental apparatus to keep the group near the apparatus. | Sept – Oct 2018 (Experiments were carried out when the birds engage in storing food for winter) | Experiments approved by Umea ethics board, A39-15. Ringing under the license of the Swedish Museum of Natural History. |
| Siberian jays (*Perisoreus infaustus*) Group 2 | Wild population, studied near Arvidsjaur, Swedish Lapland (65°40 N, 19°0 E) | *Subjects:* male and female breeder, one juvenile born spring 2018. *Living area:* See above for details. | Sept – Oct 2018 (Experiments were carried out when the birds engage in storing food for winter) | Experiments approved by Umea ethics board, A39-15. Ringing under the license of the Swedish Museum of Natural History. |

*Continued on next page*

*Appendix 2—table 1 continued*

| Species | Study site | Subject and husbandry details | Testing period | Ethical approval |
|---|---|---|---|---|
| Eurasian jackdaws (*Corvus monedula*) | Comparative Cognition Research Group of the Max-Plank-Institute for Ornithology in Seewiesen, Germany | *Subjects:* 7 males and seven females adult birds (>4 years old), most of the birds were hand-raised. one subject (male) participated only in phases 0–2; one subject (female) only participated in phases 0–3. two subjects (1 male and one female) joined the group in June 2017 and participated in phases 3–6. *Housing:* the birds had access to two aviaries (aviary 1: 15m × 9 m × 2.80 m; aviary 2: 12m × 10 m × 2.80 m) with adjacent experimental compartments. All compartments had natural soil and vegetation, including bushes and small trees, and were equipped with breeding boxes, several birdbaths, and other enrichment objects. *Feeding:* the birds were fed a diverse diet consisting of meat, insects, curd, rice, cereals and Versele Laga Nutribird Beo pearls, and fruit twice a day; water was provided ad libitum. The food was enriched with mineral and vitamin supplements. | Aug 2016 – Aug 2017; re-test: Sep 2017 | The study followed the protocols of the University of Vienna and followed the guidelines of the Association for the Study of Animal Behaviour and conformed the European and German legalisations and guidelines for the use of animals. All animals were habituated to humans. |

