## [Decision Letter]

**Acceptance summary:**

This study presents an important contribution to comparative animal cognition research. Using an experimental paradigm that was originally used to test prosociality in primates, this study tests prosociality across eight species of corvids. Of particular interest is the finding that the species-specific traits of cooperative breeding and colonial nesting effect prosocial behavioural tendencies but do so to varying degrees for males and females. This work therefore provides valuable insight into the potential evolutionary pathways and drivers of prosociality.

**Decision letter after peer review:**

Thank you for submitting your article "Sex-specific effects of cooperative breeding and colonial nesting on prosociality in corvids" for consideration by *eLife*. Your article has been reviewed by three peer reviewers, one of whom is a member of our Board of Reviewing Editors, and the evaluation has been overseen by a Reviewing Editor and Detlef Weigel as the Senior Editor. The reviewers have opted to remain anonymous.

The reviewers have discussed the reviews with one another and the Reviewing Editor has drafted this decision to help you prepare a revised submission.

As the editors have judged that your manuscript is of interest, but as described below that additional analyses are required before it is published, we would like to draw your attention to changes in our revision policy that we have made in response to COVID-19 (https://elifesciences.org/articles/57162). First, because many researchers have temporarily lost access to the labs, we will give authors as much time as they need to submit revised manuscripts. We are also offering, if you choose, to post the manuscript to bioRxiv (if it is not already there) along with this decision letter and a formal designation that the manuscript is "in revision at *eLife*". Please let us know if you would like to pursue this option. (If your work is more suitable for medRxiv, you will need to post the preprint yourself, as the mechanisms for us to do so are still in development.)

Summary:

The authors present an interesting study on the prosocial tendencies of eight bird species within the Corvid family, using species-specific predictors of colonial nesting and cooperative breeding. Importantly, the study replicates an experimental paradigm used to test prosociality in several species of primates. The authors show that cooperative breeding and colonial nesting affect prosocial behaviors, with interesting interactions by sex, apparently driven by female cooperative breeders and male colonial nesters.

Essential revisions:

The reviewers all found the comparative approach to be worthwhile and the paper to be well written and easy to follow for the most part, but all three had major concerns with the analyses. There were also some concerns with the experimental procedure, which might be addressed by simply modifying or adding analyses. The authors should also include a file with all the R code used to run and interpret the models and all other analyses (e.g, Pielou's J), as well as the source files for figures when submitting a revision.

1) It is not clear why the authors chose to apply different criteria for different parts of the analyses. In particular, passing the last criteria (test vs controls) appears to be crucial to establish prosociality in these species (and actually is a test in itself). We agree with the authors that there are reasons why birds might not have passed (e.g. as discussed lacking cooperation by receiving birds), but at the very least, we would like see the results of an analysis for just the birds that passed the criterion and how these results compare to the presented ones (as the authors did for percentage of provided food). Of course, this will drastically reduce sample size but such an analysis would be especially important considering the percentage of birds passing the criteria was high in the cooperatively breeding and/or colonial species.

2) Why were test and control sessions conducted on alternating days (rather than pseudo-randomly distributing them throughout the session, or even better on a trial level)? With the current design we are concerned that the data is not independent within a day. (The same applies, albeit to a lesser degree as it is only the habituation phase to phase I). Given that this can't be changed anymore, can it be accounted for in the models?

3) Many of the effects observed for each species are driven by very few individuals, which casts doubt on how well the results reflect true species generalizations rather than individual personalities. For example, the species where more than one group could be tested showed a lot of variation, presumably due to the presence of particular individuals. Could the authors (1) clarify in the manuscript that for all models/analyses that only one data point per individual was used? (2) Could the authors provide more discussion about possible inter-individual differences and how this could effect their results?

4) Why was a phylogenetic generalized linear mixed model (pglmm) not used, especially considering the variation in relatedness among the 8 species (seen in Figure 2)? Please provide clear justification or else re-run using a pglmm framework.

5) The model selection approach is problematic for a number of reasons. Firstly, the candidate set of models was not clear (this should be included clearly with the code for the paper) and where did the intercept only model fall relative to the others when ranked by AIC? This needs to be explicitly discussed in the Results section. Second, the p values being reported (e.g., Table 2) are not understandable, are they from two different models? An average? Why are you reporting them at all and not instead model-weighted averages (e.g., summed akaike weights) of the different predictors, including group size, considering you are using an information theory-based approach with AIC? With relatively few predictors and strong theoretical support for each, such as in this study, selecting the best models (delta AIC <2) seems arbitrary and leads to removing potentially important variables, like group size, based on this threshold (note Burnham et al., 2010 also note ' Models where Δ is in the 2-7 range have some support and should rarely be dismissed'). A more parsimonious approach is to use a model set and model weighted averages of coefficients and SEs and Akaike weights to assess covariate support (see Mundry, 2011 and Burnham et al., 2011 for best practices associated with multimodel inference and how to report results).

6) Table 2 of the main results is somewhat misleading since reporting coefficients and p values of main effects when their interaction is significant are problematic (see for example Brambor, Clark and Golder, 2006). It is fine to demonstrate via plotting the unconditional effects of the two factors, but Table 2 on its own is confusing since the interaction tells us that the main effects are conditional upon one another.

7) The Introduction suggests sex ratio of groups may be an important predictor of prosocial tendencies in some species. Considering that sex ratio varies among the social groups of birds tested, this should be included as a predictor in the models. Similarly, is there any reason to suspect variation according to age, i.e., juveniles and adults? If so, should this not also be included?

8) The sex-specific models with only 25/26 data points suggests these models may be incredibly unstable, and there is no mention of group ID or species being included. If these terms were dropped from these models this should have been tested as you did with the full data set but we could not find this mentioned anywhere. Moreover, can you provide some measure of how robust and stable the results of the sex-specific models are? For example, check how much variation there is in your coefficients if one species is removed at a time? A similar exercise would also add credibility to your results for the full data set.

[Editors' note: further revisions were suggested prior to acceptance, as described below.]

Thank you for resubmitting your work entitled "Sex-specific effects of cooperative breeding and colonial nesting on prosociality in corvids" for further consideration by *eLife*. Your revised article has been evaluated by Detlef Weigel (Senior Editor) and a Reviewing Editor.

The manuscript has been significantly improved but there are some remaining issues that need to be addressed before acceptance, as outlined below:

1) Regarding our previous major concern, point #2, we understand that day is confounded with test condition, but could the authors not simply add day as a random effect (intercept only, no slopes needed to keep the model simple)? We would like the authors to consider this approach if possible, to fit.

2) Regarding the model stability check for the single sex models and for the full model, we suggest the authors add a short description of how they did this to the section titled 'Data analysis' and when reporting the results, especially for the single sex models, explicitly state in the manuscript that the female model is less robust/stable (similar to what you wrote in your response) and that in general the single sex model results are very preliminary due to the low sample size. More data are definitely needed here.

3) We appreciate the authors checking whether they obtained similar results when using a cut off of AIC<7 for 'top models'. We would suggest that for transparency, the authors also add that they did this additional check (subsection “Data Analysis”) and add Table R1 (in the response letter) as a supplementary table. Since the results do differ slightly, we think it is worthwhile to provide the reader with all possible information.

---

## [Author Response]

Summary:The authors present an interesting study on the prosocial tendencies of eight bird species within the Corvid family, using species-specific predictors of colonial nesting and cooperative breeding. Importantly, the study replicates an experimental paradigm used to test prosociality in several species of primates. The authors show that cooperative breeding and colonial nesting affect prosocial behaviors, with interesting interactions by sex, apparently driven by female cooperative breeders and male colonial nesters.Essential revisions:The reviewers all found the comparative approach to be worthwhile and the paper to be well written and easy to follow for the most part, but all three had major concerns with the analyses. There were also some concerns with the experimental procedure, which might be addressed by simply modifying or adding analyses. The authors should also include a file with all the R code used to run and interpret the models and all other analyses (e.g,. Pielou's J), as well as the source files for figures when submitting a revision.

We created a source code file that allows readers to reproduce our calculations of Pielou’s J’ and our statistical models in R. The data used to create Figure 2 is taken from the dataset that is available in full on Dryad (Corvid_GSP_Data.csv). The data used to create Figure 3 is now available as Figure 3—source data 1.

1) It is not clear why the authors chose to apply different criteria for different parts of the analyses. In particular, passing the last criteria (test vs controls) appears to be crucial to establish prosociality in these species (and actually is a test in itself). We agree with the authors that there are reasons why birds might not have passed (e.g. as discussed lacking cooperation by receiving birds), but at the very least, we would like see the results of an analysis for just the birds that passed the criterion and how these results compare to the presented ones (as the authors did for percentage of provided food). Of course, this will drastically reduce sample size but such an analysis would be especially important considering the percentage of birds passing the criteria was high in the cooperatively breeding and/or colonial species.

We agree with the reviewers that it is a very interesting aspect to look at the landings of the individuals that reached the criterion of being “prosocial” (i.e., meeting the criterion of landing significantly more often in the test than in both controls) and whether they differ between individuals from species that are cooperative breeders (or not) or that are colonial nesters (or not). Due to the extremely small sample of significantly “prosocial” individuals (N=12), we were not able to calculate models that included both factors “cooperative breeding” and “colonial nesting” simultaneously or to include the important interactions with “sex”. But we used non-parametric tests to assess differences between the two factors “cooperative breeding” and “colonial nesting” separately. Among the “prosocial” individuals we found a non-significant trend that individuals from colonial species landed more often than individuals from territorial species (Mann-Whitney test, N=12, W=30, p=0.0505). Individuals from cooperatively breeding species did not differ in the number of their landings from individuals from species that do not breed cooperatively (Mann-Whitney test, N=12, W=13, p=0.4696). We report and discuss these results in the manuscript (subsection “Linking cooperative breeding and colonial nesting with prosocial behavior”; Discussion section).

2) Why were test and control sessions conducted on alternating days (rather than pseudo-randomly distributing them throughout the session, or even better on a trial level)? With the current design we are concerned that the data is not independent within a day. (The same applies, albeit to a lesser degree as it is only the habituation phase to phase I). Given that this can't be changed anymore, can it be accounted for in the models?

It is an important aspect of the group service paradigm that each session is composed only of trials of one condition, because the aim is for the animals to learn the contingencies of the specific condition. Intermixing trials of different conditions within one session would have placed much more cognitive load on the participating birds (e.g., attention, inhibition) and might have prevented the birds from ever learning the contingencies. Burkart et al., specifically developed this paradigm to be cognitively simple (discussed in Burkart et al., 2013), while allowing to measure the animals’ prosocial tendencies by testing whether they would continue to provide for their group members after they had learned the contingencies of each condition over 5 sessions each (i.e., that the group members, but not the individual itself obtains food in the prosocial test; that nobody can obtain food in the control conditions). Moreover, the blocked control sessions required a modification of the apparatus (i.e. installing a fine-meshed net that prevented birds on the receiving side to obtain a reward). Due to high levels of neophobia in corvids (Greenberg and Mettke-Hofmann, 2001), this modification required a pause of a few days between Phase IV (Prosocial Test and Empty Control) and Phase V (Blocked Control and Blocked Empty Control) and we could therefore not have alternated sessions with and without the fine-meshed net installed.

We mitigated the problem of having sessions of different conditions on different days by conducting them at the same testing times on each day for each respective species, thereby aiming to have comparable motivation levels (e.g., hunger) for all conditions. Our analysis of the motivation trials shows that there was no significant difference in how many food pieces were obtained in the motivation trials of each condition (see Appendix 1—table 5), which suggests that the birds’ motivation to participate in the experiment was comparable in all conditions. Unforeseeable surrounding circumstances (e.g., bad weather) happened equally often for the different conditions and did not seem to affect the birds’ motivation to participate. Below, we include graphs for the birds’ responses in the different conditions over the five sessions, to illustrate how they learned the contingencies over the testing days and how they behaved across days and between conditions.

We used the procedure of conducting test and control sessions on alternating days to keep the procedure as comparable as possible to the one designed by Burkart et al., and in order to avoid potential carry-over effects from two consecutive test or control sessions. But we agree with the reviewers that for future studies, it might be interesting to introduce a pseudo-randomized sequence or even test the birds’ cognitive abilities by intermixing trials of different conditions within one session and seeing how flexibly the birds can switch from one behavioral strategy to another.

We added more information about this issue in the main manuscript (i.e., “To ensure that the birds had comparable motivation levels (e.g., hunger) in all conditions we conducted all sessions at the same testing times per day for each respective species.” subsection “Apparatus and procedure”) and in Appendix 2 (subsection “Phase IV – Group service (Test phase)”). Since the factor “testing day” is confounded with “condition” in the current procedure, we were not sure how to include this in our statistical models.

**Author response image 1. sa2fig1:** Number of landings across the testing days of Phase IV (prosocial test vs empty control) for all groups. The box plots represent medians (horizontal lines), inter-quartile ranges (boxes), as well as minima, maxima (whiskers). Dots not encompassed by the whiskers are outliers.

**Author response image 2. sa2fig2:** Number of landings across the testing days of Phase V (blocked control vs blocked empty control) for all groups. The box plots represent medians (horizontal lines), inter-quartile ranges (boxes), as well as minima, maxima (whiskers). Dots not encompassed by the whiskers are outliers.

3) Many of the effects observed for each species are driven by very few individuals, which casts doubt on how well the results reflect true species generalizations rather than individual personalities. For example, the species where more than one group could be tested showed a lot of variation, presumably due to the presence of particular individuals. Could the authors (1) clarify in the manuscript that for all models/analyses that only one data point per individual was used? (2) Could the authors provide more discussion about possible inter-individual differences and how this could effect their results?

1) We confirm that only one data point per individual was used in all statistical information and added this remark in the subsection “Data analysis”.

2) We added a more substantial discussion about the possible inter-group differences and inter-individual differences, the potential sources of these issues, and some ideas about how to ameliorate such effects in future studies (Discussion section).

4) Why was a phylogenetic generalized linear mixed model (pglmm) not used, especially considering the variation in relatedness among the 8 species (seen in Figure 2)? Please provide clear justification or else re-run using a pglmm framework.

To test the extent to which common ancestry affected the birds’ prosocial tendencies, we added a phylogenetically controlled mixed-effects model with “number of landings in the prosocial test” as response variable, and those parameters that were present in the top 2AICc models of the original analysis (i.e., “cooperative breeding”, “nesting type”, “sex”, and the interactions between “cooperative breeding” and “sex” and “nesting type” and “sex”). Additionally, we added “phylogenetic effect” and “species” as random effects. We further calculated the posterior mean (mean of the posterior distribution), the posterior mode (most likely value regarding the posterior distribution) and the 95% credible interval of the phylogenetic signal λ. We found that the results were equivalent to the original model: the main factors cooperative breeding and sex significantly predicted the number of landings on the provisioning perch in the prosocial test, and these main effects were again qualified by significant interactions between both cooperative breeding and sex and nesting type and sex. The phylogenetic signal was weak. We report and discuss these results in the "manuscript (subsection “Testing the effect of phylogeny on prosocial behavior”; Discussion section).

5) The model selection approach is problematic for a number of reasons. Firstly, the candidate set of models was not clear (this should be included clearly with the code for the paper) and where did the intercept only model fall relative to the others when ranked by AIC? This needs to be explicitly discussed in the Results section.

We explained in more detail how we obtained the candidate set of models and that the intercept-only model did not fall within the final selection of best-fitting models (subsection “Data analysis”).

Second, the p values being reported (e.g., Table 2) are not understandable, are they from two different models? An average? Why are you reporting them at all and not instead model-weighted averages (e.g., summed akaike weights) of the different predictors, including group size, considering you are using an information theory-based approach with AIC?

We revised the table and removed the p-values and confidence intervals. We also realized that the values of the sum of AICc weights was mislabeled as “relative importance” and we corrected this error. We further included a column with the number of models in which each given factor was present. We highlighted these factors in bold, which were considered to have a high explanatory degree (i.e., factors with a sum of AICc weights larger than 0.5 and whose SE of the estimates did not overlap 0). The table is now presented as Figure 2—figure supplement 1.

With relatively few predictors and strong theoretical support for each, such as in this study, selecting the best models (delta AIC <2) seems arbitrary and leads to removing potentially important variables, like group size, based on this threshold (note Burnham et al., 2010 also note ' Models where Δ is in the 2-7 range have some support and should rarely be dismissed'). A more parsimonious approach is to use a model set and model weighted averages of coefficients and SEs and Akaike weights to assess covariate support (see Mundry, 2011 and Burnham et al., 2011 for best practices associated with multimodel inference and how to report results).

In order to ascertain that our cut-off point of delta AICc≤2 did not lead us to remove potentially important variables, we repeated the model selection and averaging procedure with delta AICc≤7. The results were equivalent to the original analysis: cooperative breeding, sex, the interaction between cooperative breeding and sex, and the interaction between nesting type and sex emerged as factors with a high explanatory degree (given in bold in Table R1). Setting the cut-off point at delta AICc≤7 only led to the inclusion of three additional factors with minimal explanatory degree (i.e., the interaction between cooperative breeding and nesting type, the factor group size, and the three-way interaction; see below). Therefore, we feel that our original approach with a cut-off point of delta AICc≤2 is supported.

Table R1 – cut-off point set to delta AICc≤7. Given are estimates, standard errors (SE), z-values, sum of AICc weights (SW_AICc_), and number of models containing the specific factor (N_Models_) after model averaging. Factors with a sum of AICc weights larger than 0.5 and whose SE of the estimates did not overlap 0 were considered to have a high explanatory degree and are given in bold.

6) Table 2 of the main results is somewhat misleading since reporting coefficients and p values of main effects when their interaction is significant are problematic (see for example Brambor, Clark and Golder, 2006). It is fine to demonstrate via plotting the unconditional effects of the two factors, but Table 2 on its own is confusing since the interaction tells us that the main effects are conditional upon one another.

Due to the revised depiction of the results we now don’t report p-values anymore. In the Results section we described in more detail that the main effects have to be viewed in the light of the important interaction terms and that they are conditional upon one another (“These main effects were qualified by the high explanatory degree of the interaction terms of both cooperative breeding and nesting type with sex (Figure 2—figure supplement 1), meaning that the main effects were conditional upon one another.”; subsection “Linking cooperative breeding and colonial nesting with prosocial behavior”) and we also refer to this issue in the Discussion section.

7) The Introduction suggests sex ratio of groups may be an important predictor of prosocial tendencies in some species. Considering that sex ratio varies among the social groups of birds tested, this should be included as a predictor in the models. Similarly, is there any reason to suspect variation according to age, i.e., juveniles and adults? If so, should this not also be included?

We are not aware of any studies suggesting that sex ratio of social groups affects prosocial tendencies in corvids or other non-human animal species. We would be happy to include such references, in case the reviewers have specific suggestions.

In order to investigate a potential effect of sex ratio on prosocial behavior, we re-ran our original analysis and added “sex ratio” as an additional predictor without interactions. We had to exclude all the Siberian jays from this analysis, since sex was unknown for a substantial number of individuals and therefore sex ratio could not be calculated. The results were very similar to the original results and the factor “sex ratio” was not present in the final selection of best-fitting models with delta AICc≤2. In order to explore the importance of the factor “sex ratio” further, we repeated the model selection and averaging procedure with delta AICc≤7. The results were equivalent to the original analysis (see Table R2). “Sex ratio” was included in this selection, but had only minimal explanatory degree (SW_AICc_=0.22; see results below). Additionally, since we had no a priori predictions about the factor “sex ratio” and because we are already suffering from low power due the low sample size, we would opt to not include the sex ratio in our manuscript.

Table R2 – including the factor “sex ratio”, cut-off point set to delta AICc≤7. Given are estimates, standard errors (SE), z-values, sum of AICc weights (SW_AICc_), and number of models containing the specific factor (N_Models_) after model averaging of all models with delta AICc≤7. Factors with a sum of AICc weights larger than 0.5 and whose SE of the estimates did not overlap 0 were considered to have a high explanatory degree and are given in bold.

Regarding the question about a possible effect of age-variation in the group: in our sample, only five groups from three species contained juvenile individuals, while six groups consisted only of adults. Therefore, our data is unfortunately not suitable for investigating this factor. However, we fully agree with the reviewers that future studies either with larger sample sizes, or where samples can be selected specifically with regard to age variation within the group, or where the same groups could be tested at different time points with differing age class ratios would be very informative regarding the question of the influence of age on prosocial behavior. We added this information in the Discussion section.

8) The sex-specific models with only 25/26 data points suggests these models may be incredibly unstable, and there is no mention of group ID or species being included. If these terms were dropped from these models this should have been tested as you did with the full data set but we could not find this mentioned anywhere.

For the datasets split by sex we used the same procedure as for the complete dataset. We first calculated linear mixed-effects models with “group ID” nested within “species” as random factors. As for the complete dataset, the variance of the random factors “group ID” and “species” was zero, resulting in a singular fit of the models and we therefore decided to calculate linear models, as we did it for the complete dataset. This information is now provided more clearly in the subsection “Data analysis”.

Moreover, can you provide some measure of how robust and stable the results of the sex-specific models are? For example, check how much variation there is in your coefficients if one species is removed at a time? A similar exercise would also add credibility to your results for the full data set.

We used the suggested procedure of removing one species at a time and ran the linear models for the complete dataset (N=51) and the two models split by sex (males, N=25; females, N=26).

Results for the complete dataset:

With this approach, 4 (out of 8) models had exactly the same results as before (i.e., “cooperation”, “sex”, “cooperation*sex”, “nesting*sex” as important parameters; removed species: Siberian jays, N_Model_=48; rooks, N_Model_=48; ravens, N_Model_; carrion crows, N_Model_=45), while in 2 models the parameter “nesting” was also revealed to be important, additionally to the originally important parameters (Removed species: New-Caledonian crows, N_Model_=46; azure-winged magpies, N_Model_=43).

In 1 model “nesting”, “sex”, “nesting*sex” were important parameters, while “cooperation” and “cooperation*sex” were only marginally important (i.e., SW_AICc_=0.44; removed species: large-billed crows, N_Model_=42)

Finally, in 1 model the null model was included in the selection of best-fitting models (Removed species: jackdaws, N_Model_=41)

Therefore, while not all models give exactly the same results as the original model, we feel that the results are consistent and show the robustness of our original results. The exception is the model without the jackdaws, where the null model was included in the selection of best-fitting models.

Results for the males:

With the exclusion of one species at a time, 7 (out of 8) models had exactly the same results as before (i.e., “nesting” as important parameter; removed species: jackdaws, N=20; Siberian jays, N=23; rooks, N=24; ravens, N=23; NC crows, N=21; LB crows, N=20; AWM, N=21).

However, in 1 model the null model was included in the selection of best-fitting models (Removed species: carrion crows, N=23)

These results are very consistent (apart from the problem with the model without the carrion crows). Also when we tested whether males from colonial species landed significantly more often than males from territorial species with a simple group comparison, there was a significant difference (Welch t-test: t = 3.01, df = 13.66, p-value = 0.005).

These analyses show the robustness of our original results with the males.

Results for the females:

With the exclusion of one species at a time, 2 (out of 8) models had exactly the same results as before (i.e., “cooperative breeding” as important parameter; removed species: Siberian jays, N=25; ravens, N=21), while in 1 model the parameter “nesting” was also revealed to be important, additionally to “cooperative breeding” (Removed species: azure-winged magpies, N=22).

However, in 5 models the null model was included in the selection of best-fitting models (Removed species: jackdaws, N=21; rooks, N=24; NC crows, N=25; LB crows, N=22; carrion crows, N=22).

Also when testing whether females from cooperatively breeding species land significantly more often than females from non-cooperatively breeding species with a simple group comparison, the difference is only marginally significant (Welch t-test: t = -1.64, df = 8.30, p-value = 0.069). Therefore, while our results are consistent with the original analysis after the removal of 3 species, the results for the females are less robust than the results for the males.

The results of this procedure thus reveal moderate to good robustness of our findings. While we acknowledge the merit of such a procedure, we are not sure if or how to report this, and would be interested in the editors’ and reviewers’ view on that.

[Editors' note: further revisions were suggested prior to acceptance, as described below.]

The manuscript has been significantly improved but there are some remaining issues that need to be addressed before acceptance, as outlined below:1) Regarding our previous major concern, point #2, we understand that day is confounded with test condition, but could the authors not simply add day as a random effect (intercept only, no slopes needed to keep the model simple)? We would like the authors to consider this approach if possible, to fit.

We appreciate the editors’ concerns about this point. However, the dependent variable in our models was the sum of the number of landings in prosocial test sessions 4 and 5. Therefore, the data from two testing days added together was entered in the models and those testing days were the same for all members of each tested group of birds. Data from the empty sessions, which were conducted on different days, were not added in these models at all. Data collection was also conducted at very different times for the different species (between April 2015 and October 2018; see Appendix 2—table 1), making testing days between species hard to compare. Therefore, we feel that it is not practicable to add a random effect “testing day” to our models. But in order to make it clearer to the readers that the summed data from the last two sessions was entered as the dependent variable in our models, we added the following information to the manuscript:

Subsection “Data analysis”: “we used only the summed data from the last two sessions (session 4 and 5) of each condition”.

Subsection “Data analysis”: “to further investigate the influence of cooperative breeding and colonial nesting on prosocial tendencies, we used the sum of the number of landings in the last two sessions of the prosocial test (phase IV) of all birds that passed the training criterion in the preceding phase”.

2) Regarding the model stability check for the single sex models and for the full model, we suggest the authors add a short description of how they did this to the section titled 'Data analysis' and when reporting the results, especially for the single sex models, explicitly state in the manuscript that the female model is less robust/stable (similar to what you wrote in your response) and that in general the single sex model results are very preliminary due to the low sample size. More data are definitely needed here.

We added the details from the model stability check in the Results section, the Discussion section, subsection “Data analysis” and Appendix 1. We specifically state that more data are needed in the Discussion section.

3) We appreciate the authors checking whether they obtained similar results when using a cut off of AIC<7 for 'top models'. We would suggest that for transparency, the authors also add that they did this additional check (subsection “Data Analysis”) and add Table R1 (in the response letter) as a supplementary table. Since the results do differ slightly, we think it is worthwhile to provide the reader with all possible information.

We added the information about the procedure with the threshold set to delta AICc≤7 to subsection “Data analysis” and added the table as Appendix 1—table 3.